# Differences in the functional brain architecture of sustained attention and working memory in youth and adults

**Omid Kardan**[1,2]*, **Andrew J. Stier**[1], **Carlos Cardenas-Iniguez**[1], **Kathryn E. Schertz**[1], **Julia C. Pruin**[1], **Yuting Deng**[1], **Taylor Chamberlain**[1], **Wesley J. Meredith**[3], **Xihan Zhang**[1,4], **Jillian E. Bowman**[1], **Tanvi Lakhtakia**[1], **Lucy Tindel**[1], **Emily W. Avery**[4], **Qi Lin**[4], **Kwangsun Yoo**[4], **Marvin M. Chun**[4], **Marc G. Berman**[1], **Monica D. Rosenberg**[1]*

**1** University of Chicago, Chicago, Illinois, United States of America, **2** University of Michigan, Ann Arbor, Michigan, United States of America, **3** University of California, Los Angeles, California, United States of America, **4** Yale University, New Haven, Connecticut, United States of America

* omidk@med.umich.edu (OK); mdrosenberg@uchicago.edu (MDR)

**Data Availability Statement:** All ABCD curated data are available at https://nda.nih.gov/edit_collection.html?id=2573. The sustained attention

## Abstract

Sustained attention (SA) and working memory (WM) are critical processes, but the brain networks supporting these abilities in development are unknown. We characterized the functional brain architecture of SA and WM in 9- to 11-year-old children and adults. First, we found that adult network predictors of SA generalized to predict individual differences and fluctuations in SA in youth. A WM model predicted WM performance both across and within children—and captured individual differences in later recognition memory—but underperformed in youth relative to adults. We next characterized functional connections differentially related to SA and WM in youth compared to adults. Results revealed 2 network configurations: a dominant architecture predicting performance in both age groups and a secondary architecture, more prominent for WM than SA, predicting performance in each age group differently. Thus, functional connectivity (FC) predicts SA and WM in youth, with networks predicting WM performance differing more between youths and adults than those predicting SA.

## Introduction

Maintaining focus over time and information in working memory—related but separable functions [1–5]—are foundational cognitive processes critical for successfully performing everyday activities across the lifespan. In addition to being integral to everyday life, these cognitive processes vary greatly across individuals [6–8] and fluctuate over time within the same person [9]. These inter- and intra-individual differences are particularly important to study in development because of their consequences for life-long achievements. For example, research in children and adolescents has suggested that attention is more predictive of later academic achievement than more general problem behaviors (e.g., aggression and noncompliance) and interpersonal skills [10–12].

network masks are available at https://github.com/
monicadrosenberg/Rosenberg_PNAS2020.
Specific data used in the analyses of the current
study are shared in the National Institute of Mental
Health Data Archive (NDA) study 1849 with doi: 10.
15154/1528288. Users with an NDA account and
approved Data Use Agreement can download the
shared data. HCP data are available at https://db.
humanconnectome.org. Analysis scripts to
generate results and figures in the manuscript are
available at https://github.com/okardan/ABCD_SA-
WM.

**Funding:** OK was supported by National Institute
on Alcohol Abuse and Alcoholism T32 AA007477.
This research was supported by the National
Science Foundation (BCS-2043740 to MDR,
S&CC-1952050 to MGB, DGE-1746045 to KES,
BCS-1558497 to MMC), National Institutes of
Health (MH 108591 to MMC), and the University of
Chicago Micro-Metcalf Program to YD and LT. The
funders had no role in study design, data collection
and analysis, decision to publish, or preparation of
the manuscript.

**Competing interests:** The authors have declared
that no competing interests exist.

**Abbreviations:** ABCD, Adolescent Brain Cognitive
Development; ACS, American Census Study;
BOLD, blood-oxygen-level-dependent; CPT,
continuous performance task; DMN, default mode
network; FC, functional connectivity; FD, frame
displacement; fMRI, functional magnetic
resonance imaging; HCP, Human Connectome
Project; LV, latent variable; PC, principal
component; PLS, partial least squares; SA,
sustained attention; SVD, singular value
decomposition; WM, working memory.

Network neuroscience proposes that cognitive and attentional processes are emergent properties of interactions between brain regions [13]. The success of recent work predicting behavior based on functional magnetic resonance imaging (fMRI) functional connectivity (i.e., the correlation between synchronous blood-oxygen-level-dependent (BOLD) activity among pairs of brain regions) supports the tenability of this position [14–18]. In other words, this work suggests that the degree to which activity is coordinated across large-scale brain networks may better characterize cognitive processes than the magnitude of activity in single regions in isolation [19].

Therefore, in the current study, we aimed to understand the development of sustained attention (SA) and working memory (WM) through the lens of network neuroscience. To do this, we used 2 approaches. First, we assessed the degree to which connectome-based predictive models of SA and WM defined in adults generalize to predict SA and WM in preadolescents. Second, we characterized the functional brain connections that are differentially related to SA and WM performance in preadolescents compared to adults, both within the constraints of the adult networks and in a whole-brain data-driven manner.

## Adult-defined connectome-based approach

Despite the popularity of connectome-based predictive modeling of behavior, cross-dataset and cross-population testing is rare. In other words, brain-based predictive models defined in 1 dataset are rarely validated in other samples, and even less so in other participant populations (e.g., different ages or diagnoses, see [20]). Hence, existing "publication preregistered" brain markers are currently underutilized and under scrutinized, which obscures both their potential and limitations [21]. Testing the generalizability of connectivity-based models across ages can inform the degree to which adults and children share common network predictors of cognition and delineate models' predictive boundaries. Cross-age model validation may also provide insight into how networks underlying cognitive and attentional processes change with development. Additionally, validating models of different cognitive processes (e.g., sustained attention and working memory) to evaluate their unique contributions to predicting behavior can determine if distinctions between the models are behaviorally relevant (e.g. see [22]) and whether those distinctions generalize to different stages of development.

To address these gaps, in Study 1, we utilized previously developed neuromarkers in the form of large-scale functional networks defined to predict SA [23] and WM [24] in adults. We applied these adult connectome-based models to data from the Adolescent Brain Cognitive Development (ABCD) Study to predict individual differences and block-to-block changes in SA and WM task performance in youth. In addition, to characterize relationships between sustained attention, working memory, and long-term memory, we asked whether these same models not only predicted ongoing task performance, but also predict subsequent recognition memory for task stimuli. Successful model generalization would suggest that the functional networks underlying SA and WM overlap between children and adults. Furthermore, a dissociation such that the neuromarker of sustained attention captures sustained attention performance, whereas the neuromarker of working memory captures working memory performance would provide evidence that separable networks support these processes in development.

## Connectome comparison approach

In the second approach, we aimed to characterize the functional connections that were differentially related to SA and WM performance in children compared to adults. This was performed both within the constraints of the adult networks and in a whole-brain data-driven manner. This approach can complement both Study 1 as well as existing work that has

revealed, for example, changes in the coupling of structural and functional connectivity (FC) profiles that may support improvements in working memory and executive abilities in adolescence [25].

To achieve this, in Study 2, we combined the behavioral and fMRI data from the youth sample ABCD Study (9 to 11 years old) and a large adult sample from the Human Connectome Project (HCP; 21 to 36+ years old). We first investigated the developmental differences in SA within the constraints of adult networks by (a) benchmarking the adult models' fit to novel adults compared to the preadolescents; and (b) computationally lesioning different components of the adult networks and comparing how much different regions of the networks were contributing to SA in the youths versus novel adults. Second, we used a multivariate method to find the set of connections that differentiated the youth and adult connectomes with regards to SA and WM performances.

## Results

### Study 1 overview

In Study 1, we tested the generalizability of network models previously defined to predict SA and WM in adults to children.

In **Study 1.1**, we asked whether the degree to which children expressed FC markers of SA [23] and WM [24] previously defined in adult data during an in-scanner *n*-back task predicted their task performance (**Fig 1**). We hypothesized that the SA connectome-based predictive

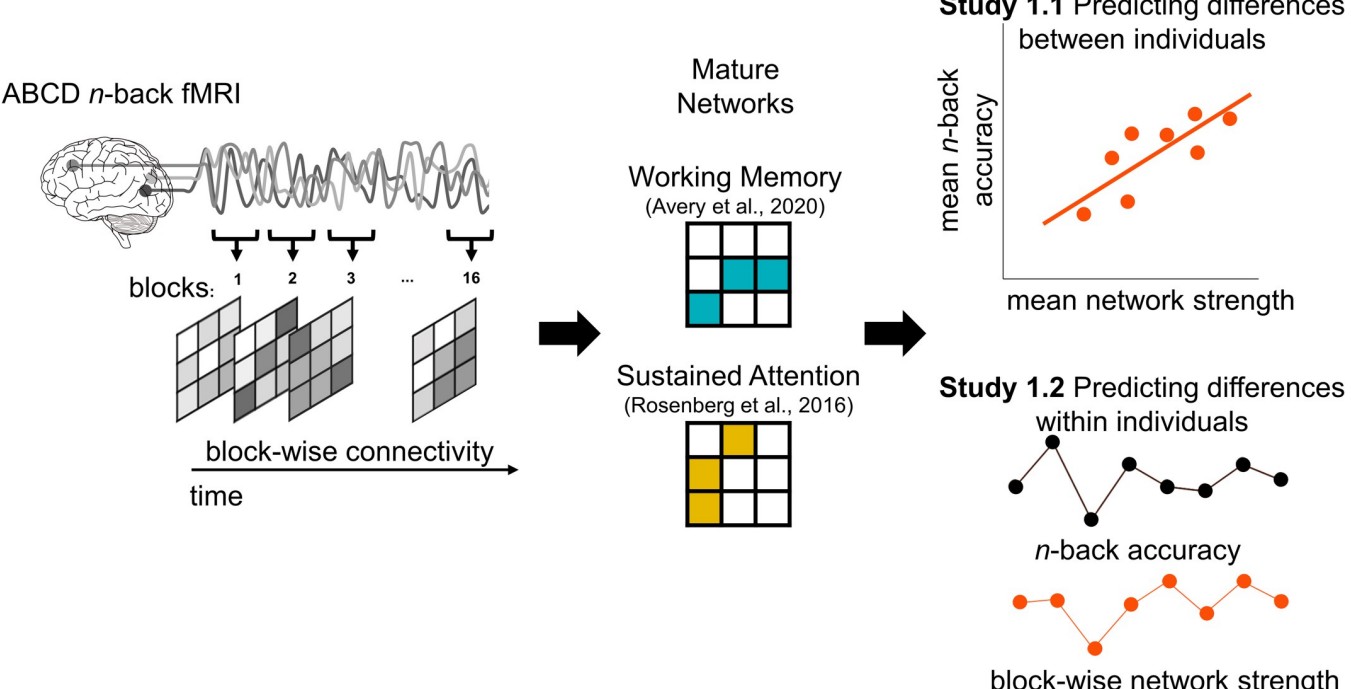

**Fig 1. Overview of Study 1.** First, we constructed block-wise FC by correlating BOLD signal time series from all pairs of functional parcels (left). For each participant, we calculated whole-brain FC patterns from fMRI data collected during the eight 0-back and eight 2-back tasks blocks. That is, we calculated up to 16 FC matrices per individual: 1 using data from each 25-s (30–31 volumes) *n*-back block separately. Each of the 2 predefined predictive network masks were then applied to each of these matrices to generate block-specific WM or SA network strength scores (middle). Each child's mean network strength during 0-back and 2-back blocks was compared to their mean accuracy in 0-back and 2-back blocks (Study 1.1) or their mean out-of-scanner recognition memory for *n*-back stimuli (Study 1.3). In Study 1.2, block-to-block changes in network strength were compared to corresponding block-to-block changes in 0-back and 2-back accuracy within-subjects. ABCD, Adolescent Brain Cognitive Development; BOLD, blood-oxygen-level-dependent; fMRI, functional magnetic resonance imaging; SA, sustained attention; WM, working memory.

model would predict 0-back task performance because this low-working-memory-load task is essentially a target detection task similar to a continuous performance task (CPT) traditionally used to assess SA (e.g., [26]). The SA network may or may not predict 2-back task performance: Although working memory and attention fluctuate in tandem in adults [27], sustained attention is not sufficient for successful 2-back task performance. We hypothesized that the WM connectome-based predictive model, on the other hand, would predict 2-back performance, and that model predictions would be more closely related to 2-back than to 0-back performance because successful 2-back (but not 0-back) performance requires the continuous maintenance and updating of items in working memory.

In **Study 1.2**, we asked whether these same adult-defined network models captured changes in SA and WM over time in children—i.e., whether block-to-block changes in network strength predicted block-to-block changes in *n*-back task accuracy (**Fig 1**, right panel). Again, we tested for specificity, asking whether the sustained attention and working memory networks better predicted sustained attention (0-back) and working memory (2-back) performance fluctuations, respectively.

Finally, in **Study 1.3**, we asked: Does the degree to which an individual shows a FC signature of better sustained attention or working memory only affect concurrent task performance, or does it also impact later cognitive processes, such as long-term memory? To investigate this question, we evaluated the consequences of SA and WM network expression for long-term memory by testing whether network strength during the *n*-back task predicted post-scan recognition memory for task stimuli.

## Study 1.1. Predicting sustained attention and working memory across participants

Do functional network models defined to predict SA and WM in adulthood generalize to a large, heterogeneous developmental sample to predict individual differences in these abilities? To test this possibility, we applied our adult connectome-based models of SA and WM to functional connectivity observed during 9- to 11-year-olds' performance of 2 *n*-back task conditions.

### Relationship between sustained attention and working memory networks

We hypothesized that the predefined SA [23] and WM [24] network masks (**Fig 2**) capture related but distinct aspects of cognitive function (see Methods for descriptions of these networks). Prior to predicting behavioral performance in the ABCD Study sample, we assessed this hypothesis by (a) comparing the anatomy of the SA and WM network masks; and (b) comparing the strength of the SA and WM networks across participants in the ABCD Study sample.

First, we found that although the SA and WM networks both include functional connections, or edges, representing coordinated activity across distributed brain regions, they show little overlap. Thirty-seven edges are common to the high-attention and high-working-memory networks (1.5% of combined edges in high-attention and high-working-memory networks, hypergeometric $p = 0.351$; see Methods), which predict better SA and WM performance, respectively. Thirty-three edges are common to the low-attention and low-working-memory networks (1.8% of combined edges, hypergeometric $p = 0.005$), which predict worse SA and WM performance, respectively. Most of these common edges involved prefrontal (32%), motor (21%), and temporal (16%) regions in the high-attention and high-working-memory networks; and cerebellar (45%), occipital (18%), and parietal (18%) regions in the low-attention and low-working-memory networks. There is no significant overlap between

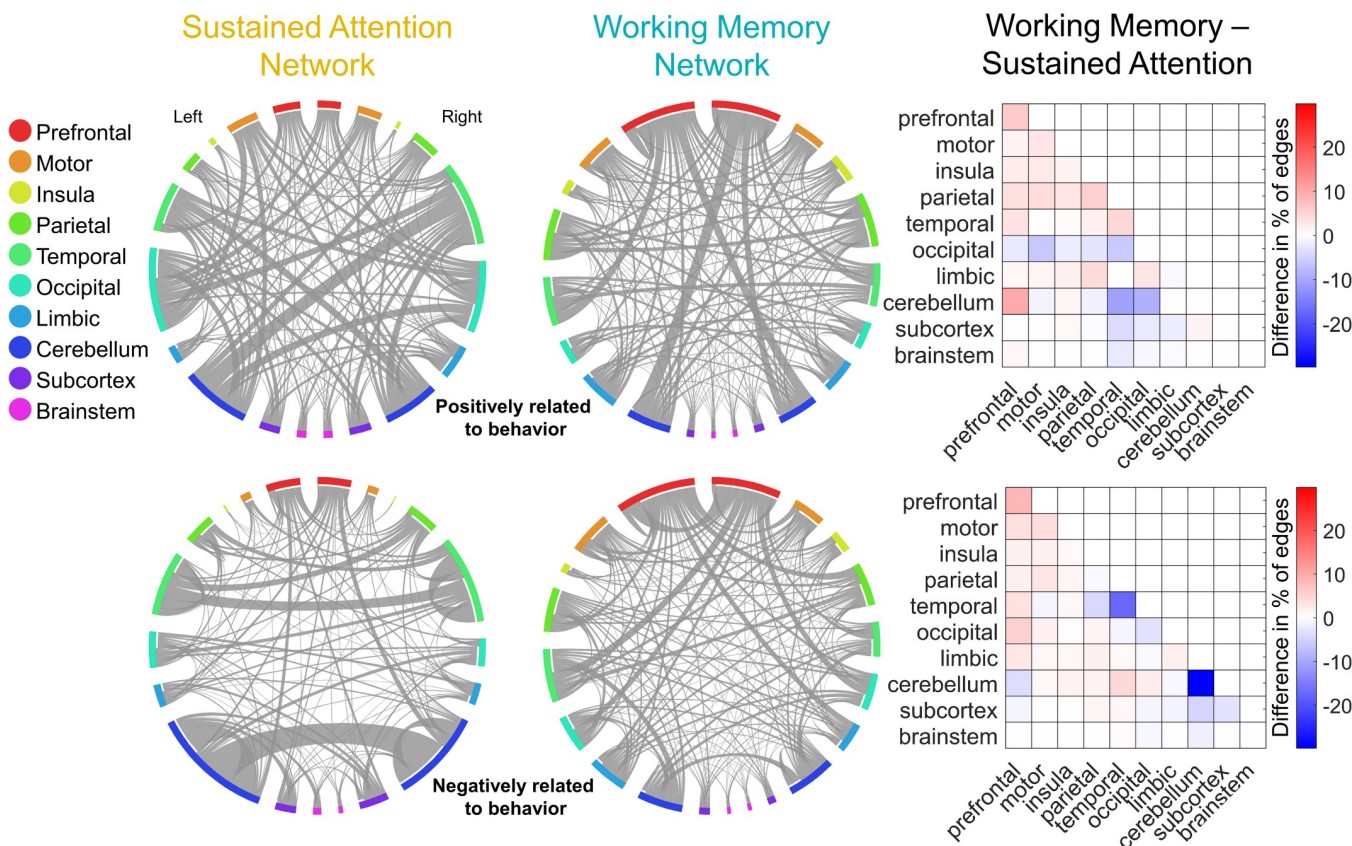

**Fig 2. Adult SA and WM networks and their differences.** The circle plots (connectograms) show the SA and WM networks on the Shen-268 parcels grouped into 20 anatomical regions (10 per hemisphere). The networks positively related to the behavior are shown on the top row and the networks negatively related to the behavior are shown in the bottom row. The matrix plots show the percentage of edges belonging to each macroscale region in WM connectogram minus the percentage of edges belonging to each macroscale region in SA connectogram. Differences in networks predicting better attention and working memory are shown in the top matrix plot; differences in networks predicting worse attention and working memory are shown in the bottom matrix plot. The data for this figure are available at NDA study 1849 10.15154/1528288. Circular plots are made with Circos [28]. SA, sustained attention; WM, working memory.

the high-attention and low-working-memory networks (19 edges, 0.9%, $p = 0.89$) or the low-attention and high-working-memory networks (12 edges, 0.5%, $p = 0.99$). At the macroscale region level, the SA networks are more dominated by cerebellar, temporal, and occipital connections, whereas the WM networks include more prefrontal connections (**Fig 2**).

Anatomical differences between the SA and WM networks, however, do not guarantee that their strength does not covary together across participants. That is, the degree to which an individual expresses the networks may not be independent. As such, we correlated SA and WM network strength during the 0-back and 2-back tasks in the ABCD Study sample (see Methods). Briefly, in the 0-back task, children were instructed to detect a target image, shown in the beginning of the block, among a series of images by pressing index versus middle finger on the response box. In the 2-back task, children saw a series of images and determined if the image in each trial matched that of 2 trials prior to it or not, again by pressing middle versus index finger. In each block, images from 1 of 4 categories: faces with positive, negative, and neutral expressions and scenes were used in the task.

Results revealed that SA and WM network strength values were positively correlated across children during the 0-back task ($r = 0.16$, $p_{adj} < 0.001$), but negatively correlated during the 2-back task ($r = -0.11$, $p_{adj} < 0.001$; black scatterplots in **Fig 3**). We found a similar pattern of

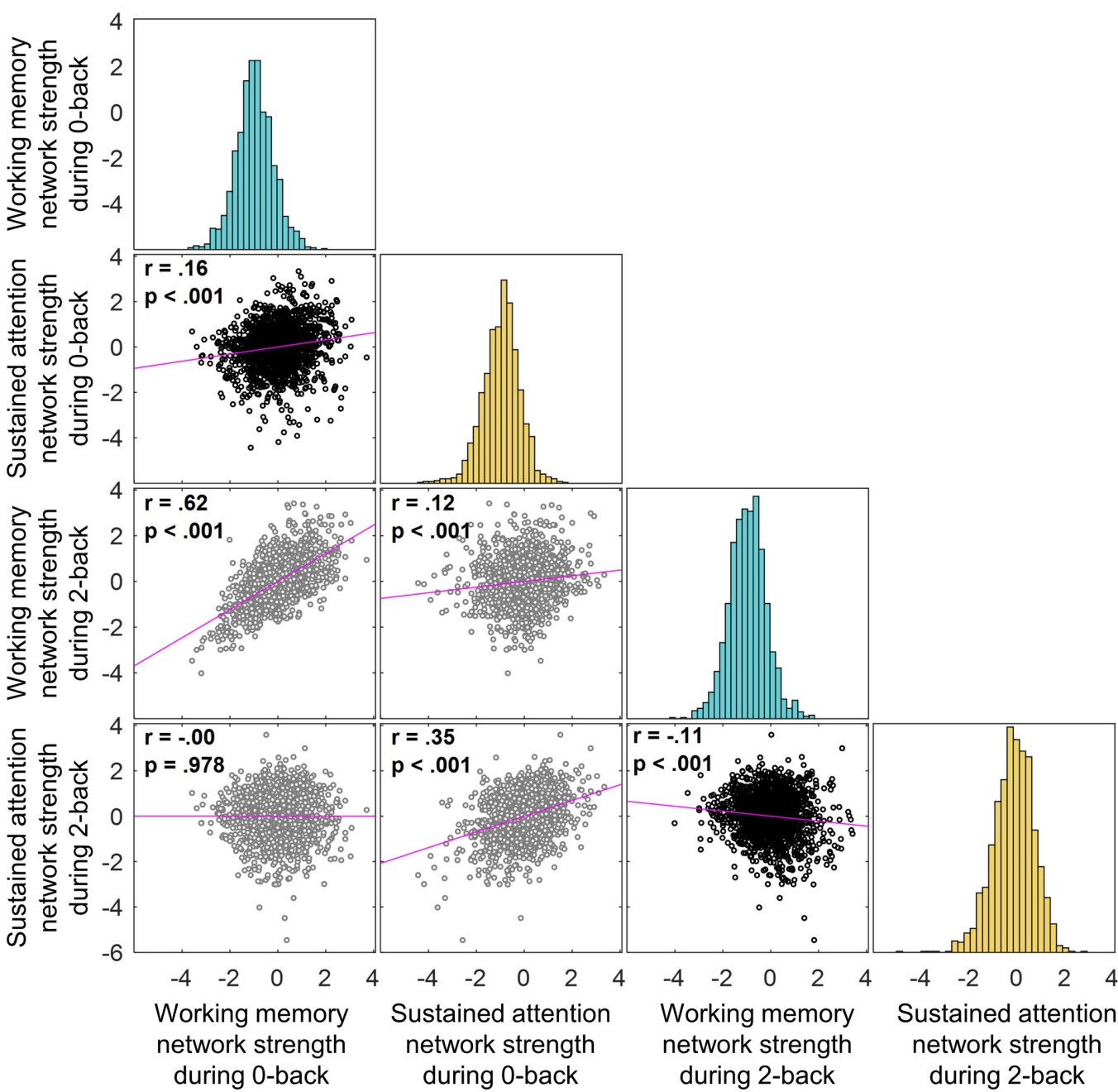

**Fig 3. Network strengths across the participants and tasks.** Correlations between predictive networks strength values across the participants in the 2-back task and 0-back tasks. Task-congruent relationships are shown in black scatterplots. The data for this figure are available at NDA study 1849 10.15154/1528288.

results within participants, such that SA and WM network strength values were positively correlated across 0-back task blocks (mean $r = 0.05$, CI = [0.02, 0.07], $p < 0.001$) and negatively correlated across 2-back task blocks (mean $r = -0.04$, CI = [−0.02, −07], $p = 0.001$). Taken together, the anatomical overlap and network strength correlation analyses both across and within participants suggest that the SA and WM masks are separable functional networks in children and thus likely do not reflect a monolithic cognitive process.

## Neuromarkers differentially predict sustained attention and working memory abilities

After confirming that the SA and WM networks are separable in children, we asked whether they generalize to specifically predict these abilities in children. To answer this question, we related adult SA and WM network strength values to task performance during 0-back and 2-back task blocks across the 9- to 11-year-old participants. Again, we predicted that the SA model would capture 0-back performance, whereas the WM model would capture 2-back performance.

Supporting our hypothesis, we found that strength of the adult SA network predicted 0-back performance ($r = 0.19$, $\rho = 0.15$, $p_{adj} < 0.001$) and strength of the adult WM network predicted 2-back performance ($r = 0.13$, $\rho = 0.14$, $p_{adj} < 0.001$) in the preadolescent youth. This external validation demonstrates cross-dataset and cross-age generalizability of the SA and WM connectome-based predictive models (**Fig 4**). This result suggests that the FC features that predict individual differences in sustained attentional and working memory abilities in adults are present and predictive in 9- to 11-year-olds.

Finally, we adjusted for data exclusions on household income, age, maternal education, race/ethnicity, sex, household size, and mean Child Behavior Checklist scores using non-participation sensitivity analysis (see Methods). The corrected correlation between WM network strength and 2-back and 0-back accuracy ($r_{corrected} = 0.134$ and $r_{corrected} = 0.115$, respectively; $ps < 0.001$), as well as between SA network strength and 0-back accuracy ($r_{corrected} = 0.246$, $p < 0.001$) were significant. Thus, these correlations are robust to the application of post-stratification weights (to account for differences between the full ABCD Study sample and nationally representative sociodemographics) and non-participation weights (to account for differences between the full ABCD Study sample and the 1,545 participants included our analyses with respect to representation of sociodemographics and psychopathology).

To assess model specificity, we compared the predictive power of the SA and WM networks for 0-back and 2-back accuracy. SA network strength was not significantly related to 2-back accuracy ($r = -0.04$, $\rho = -0.07$, $p = 0.11$). This correlation was significantly weaker than the correlation between WM network strength and 2-back accuracy (William's $t$ [test of difference between 2 dependent correlations sharing 1 variable] $= -4.65$, $p < 0.001$). Thus, the WM network was a better predictor of performance on the high-working-memory load 2-back task than the SA network. We did not observe this dissociation for the 0-back task accuracy. Instead, WM network strength predicted 0-back accuracy ($r = 0.15$, $\rho = 0.14$, $p_{adj} < 0.001$), and this correlation was numerically but not significantly lower than the correlation between SA network strength and 0-back accuracy (William's $t = -1.27$, $p = 0.20$). SA network strength was more correlated with 0-back accuracy than it was with 2-back accuracy (William's $t = 10.93$, $p < 0.001$), but the WM strength was not more predictive of 2-back than it was of 0-back accuracy (William's $t = -0.78$, $p = 0.44$).

Strength in the SA and WM networks was correlated across children ($r = 0.16$, $p < 0.001$ during 0-back; $r = -0.11$, $p < 0.001$ during 2-back; **Fig 3**), and performance in 0-back and 2-back tasks are typically correlated across individuals ($r = 0.62$, $p < 0.001$ in the current sample of 1,545 children). Thus, it is important to further assess the unique contributions of the SA and WM networks to 0-back and 2-back task performance. To this end, we included both SA and WM network strength in a regression model to predict either 0-back or 2-back accuracy (**Table 1**). The regression also included age, sex, and remaining head motion (after exclusion, see Methods) as covariates, as well as random intercepts for data collection sites. Echoing the correlation results, SA network strength predicted 0-back accuracy better than chance ($\beta = 0.16$, t $= 6.35$, $p < 0.001$) and better than it predicted 2-back performance ($p < 0.001$ based on

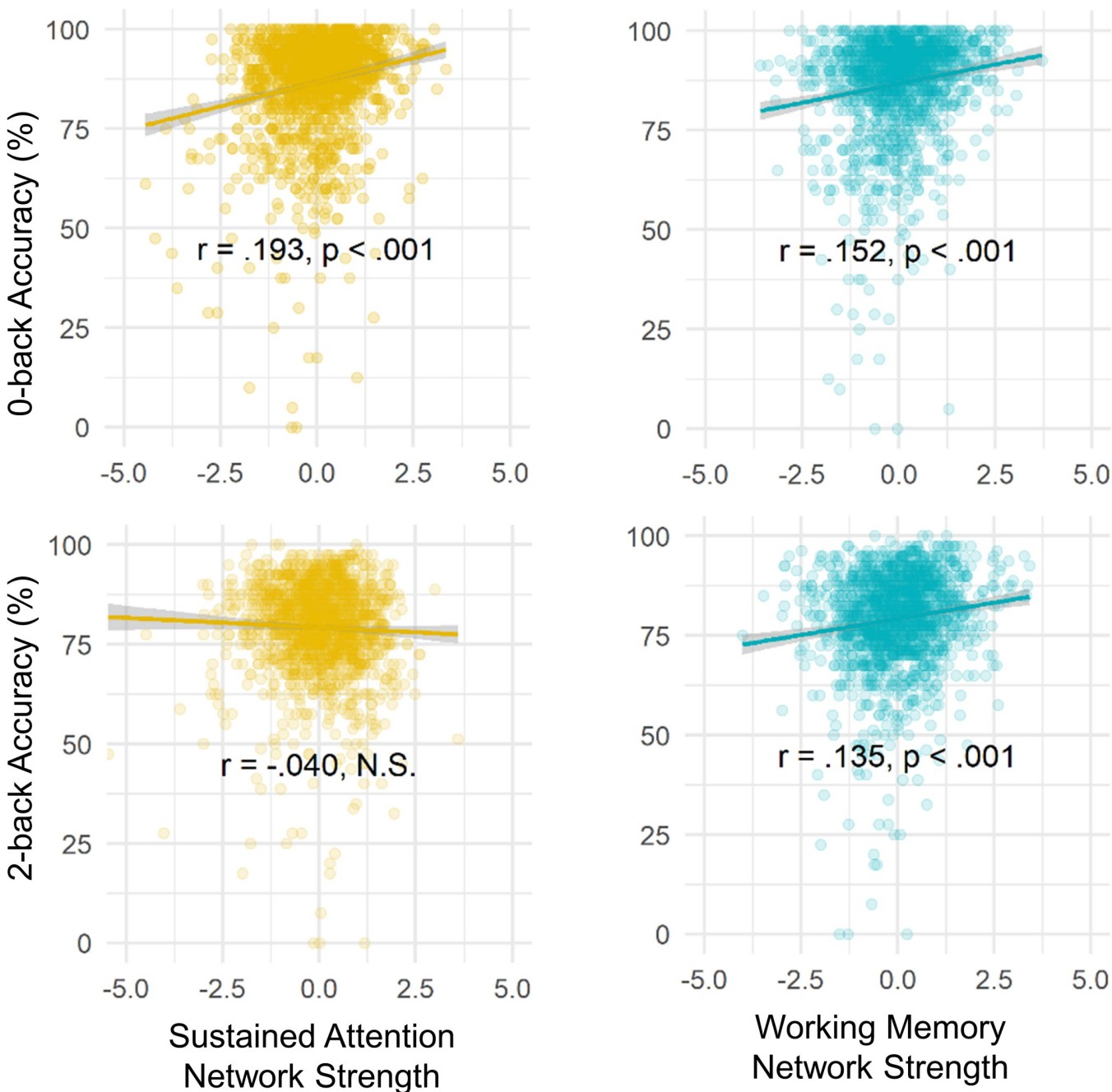

**Fig 4. Strength of adult SA and WM networks in preadolescents predict respective task performance.** Correlations between SA (left, golden) and WM (right, blue) network strength and children's 0-back (top) and 2-back (bottom) task performance. The data for this figure are available at NDA study 1849 10. 15154/1528288. SA, sustained attention; WM, working memory.

bootstrapped distribution of the difference between $\beta$ coefficients). In contrast, WM network strength predicted 0-back and 2-back accuracy above chance but equally well ($\beta = 0.11$, t = 4.19, and $\beta = 0.10$, t = 3.89, respectively; $p$ values < 0.001). Therefore, we found partial support for the specificity of the models, such that the SA network predicts 0-back accuracy better than it predicts 2-back accuracy, whereas the WM network predicts both 0-back and 2-back accuracy.

**Table 1. Strengths of both SA and WM networks as predictors of task performance.**

| Predictors | 0-back Accuracy | | | | | 2-back Accuracy | | | | |
|---|---|---|---|---|---|---|---|---|---|---|
| | Beta | CI | t | p | $pr^2$% | Beta | CI | t | p | $pr^2$% |
| (Intercept) | 0.04 | [−0.07, 0.12] | 0.67 | 0.513 | _ | 0.13 | [0.02, 0.24] | 2.36 | 0.026 | _ |
| Age | 0.14 | [0.09, 0.19] | 5.56 | **<0.001** | 1.87 | 0.15 | [0.11, 0.20] | 6.21 | **<0.001** | 2.31 |
| Female | −0.08 | [−0.18, 0.02] | −1.65 | 0.099 | 0.17 | −0.25 | [−0.34, −0.15] | −5.03 | **<0.001** | 1.51 |
| Motion | −0.04 | [−0.09, 0.01] | −1.59 | 0.112 | 0.12 | −0.13 | [−0.18, −0.08] | −5.28 | **<0.001** | 1.63 |
| SA network strength | 0.16 | [0.11, 0.21] | 6.35 | **<0.001** | 2.42 | −0.04 | [−0.09, 0.00] | −1.77 | 0.078 | 0.18 |
| WM network strength | 0.11 | [0.06, 0.16] | 4.19 | **<0.001** | 1.09 | 0.10 | [0.05, 0.15] | 3.89 | **<0.001** | 0.84 |
| **Random effects (site)** | | | | | | | | | | |
| ICC | $0.03_{site}$ | | | | | $0.03_{site}$ | | | | |
| Observations | 1,545 | | | | | 1,545 | | | | |
| Marginal $R^2$/conditional $R^2$ | 0.074/0.102 | | | | | 0.079/0.105 | | | | |

Individual differences in SA and WM network strength are differentially related to individual differences in children's 0-back (left) and 2-back (right) performance, respectively. "Motion" is mean frame-to-frame displacement during 0-back (left) or 2-back (right) blocks. "Sustained Attention" and "Working Memory" are mean SA network strength and mean WM network strength during 0-back (left) or 2-back (right) blocks. Marginal and conditional $R^2$ statistics estimate fixed-effects $R^2$ and total (i.e., fixed + random effects) $R^2$, respectively, based on [29]. Marginal semi-partial $r^2$ is calculated for each variable in the regression ($pr^2$%) and shown as percentage of total variance uniquely explained by the predictor using package partR2 in R.

SA, sustained attention; WM, working memory.

Two factors may contribute to the lack of specificity of the WM connecotme-based model. First, the 0-back task does require memory for the target image introduced at the start of each 0-back block, and thus is a low-load rather than a no-load task. Second, the adult WM model was originally defined to predict individual differences in 2-back task performance from adult connectomes comprised of both 0-back and 2-back fMRI data in the HCP sample [24], potentially increasing its sensitivity to *n*-back task performance overall. Future work assessing the SA and WM models' generalizability to different datasets and behavioral measures will further inform their sensitivity and specificity.

Finally, we compared the performance of the SA and WM models to that of 8 canonical functional networks (the medial-prefrontal, frontoparietal, default, subcortical-cerebellar, motor, visual I, visual II, and visual-association networks; [30]). To do so, we regressed *n*-back accuracy across participants on the average strength of all within-network connections in each network, with age, sex, and motion as covariates and site as random intercept (similar to **Table 1**). The SA and WM networks significantly outperformed all 8 canonical networks when predicting 0-back and 2-back accuracy, respectively (Z-test between $\beta$ coefficients of SA network against each of the 8 canonical networks: $Zs > 4.74$, $p$ values $< 0.001$ for 0-back accuracy and $Zs > 2.39$, $p$ values $< 0.017$ for the WM network and 2-back accuracy). Furthermore, the SA and WM networks also significantly outperformed randomly selected size-matched sets of connections from outside these networks for predicting 0-back and 2-back accuracy, respectively ($p$ values $< 1/200$ for both networks when compared to 200 random networks).

## Study 1.2. Tracking changes in sustained attention and working memory over time

Do the adult connectome-based models of sustained attention and working memory also vary with children's performance fluctuations? To test this, we examined relationships between block-to-block fluctuations in network strength and block-to-block fluctuations in task

**Table 2. Block-by-block networks strength and task performance.**

| Predictors | 0-back Accuracy | | | | | 2-back Accuracy | | | | |
|---|---|---|---|---|---|---|---|---|---|---|
| | Beta | CI | t | p | pr²% | Beta | CI | t | p | pr²% |
| (Intercept) | 0.07 | [0.02, 0.12] | 2.93 | **0.003** | _ | 0.03 | [−0.01, 0.08] | 1.37 | 0.172 | _ |
| Block motion | −0.03 | [−0.05, −0.01] | −3.06 | **0.002** | 0.13 | −0.02 | [−0.04, −0.00] | −2.26 | **0.024** | 0.07 |
| Block type: | | | | | 0.88 | | | | | 2.21 |
| Negative face | −0.13 | [−0.18, −0.09] | −5.74 | **<0.001** | _ | −0.00 | [−0.05, 0.04] | −0.16 | 0.876 | _ |
| Positive face | 0.01 | [−0.03, 0.06] | 0.64 | 0.522 | _ | −0.03 | [−0.07, 0.02] | −1.16 | 0.245 | _ |
| Place | −0.22 | [−0.26, −0.17] | −9.51 | **<0.001** | _ | −0.36 | [−0.40, −0.31] | −15.60 | **<0.001** | _ |
| Run 2 −Run 1 | 0.02 | [−0.02, 0.05] | 0.80 | 0.424 | 0.01 | 0.15 | [0.11, 0.18] | 7.75 | **<0.001** | 0.53 |
| Block SA network strength | 0.07 | [0.05, 0.09] | 7.37 | **<0.001** | 0.30 | −0.01 | [−0.03, 0.00] | −1.46 | 0.145 | 0.02 |
| Block WM network strength | 0.05 | [0.03, 0.07] | 5.56 | **<0.001** | 0.49 | 0.04 | [0.02, 0.06] | 3.83 | **<0.001** | 0.14 |
| **Random effects (subject)** | | | | | | | | | | |
| ICC | 0.40$_{subs}$ | | | | | 0.39$_{subs}$ | | | | |
| Observations (blocks) | 9,176 | | | | | 9,176 | | | | |
| Marginal R²/conditional R² | 0.019/0.412 | | | | | 0.030/0.411 | | | | |

Block-by-block changes in SA and WM networks strength values are differentially related to block level 0-back and 2-back performance in children, respectively. Marginal and conditional $R^2$ statistics estimate fixed-effects $R^2$ and total (i.e., fixed + random effects) $R^2$, respectively, based on [29]. Marginal semi-partial $r^2$ is calculated for each variable in the regression ($pr^2$%) and shown as percentage of total variance uniquely explained by the predictor using package partR2 in R. SA, sustained attention; WM, working memory.

performance within-participants. We also investigated whether changes in network strength and behavior were driven by stimulus types or were more spontaneous.

Mixed-effects block-level regressions with random intercepts for participants (**Table 2**) showed that block-by-block changes in SA network strength tracked block-to-block fluctuations in 0-back accuracy ($\beta = 0.07$, t = 7.37, $p < 0.001$) and block-to-block fluctuations in WM network strength tracked block-by-block 2-back accuracy ($\beta = 0.04$, t = 3.83, $p < 0.001$). These results were consistent with our predictions in Study 1.1. Again, demonstrating partial specificity, adult SA network strength did not significantly track 2-back accuracy ($\beta = -0.01$, t = $-1.46$ $p = 0.015$), whereas adult WM network strength did track 0-back accuracy ($\beta = 0.05$, t = 5.56, $p < 0.001$) in youth.

The observed relationships between functional network strength and task accuracy are above and beyond the variance in block-by-block $n$-back accuracy explained by potential practice effects (i.e., run 2 versus run 1) or stimulus type (i.e., positive versus neutral faces, negative versus neutral faces, and places versus neutral faces; see Tables A and B in S1 Text) because these potential sources of variance are included as covariates in the regression model. Despite the numerically small effect sizes, it is noteworthy that the strength of SA and WM networks—developed in completely independent datasets to predict individual differences in <u>adults</u>—track within-person fluctuations in 0-back and 2-back accuracy in <u>children</u>.

Similar to the across-participant results, we tested the SA and WM networks against the 8 canonical networks and 200 random size-matched networks in regressions with block motion, stimulus type, and run as covariates and subjects as random intercept (similar to Table 2). SA network strength was significantly more predictive of block-wise 0-back performance than any of the 8 canonical networks (comparison of beta coefficients: $Zs > 6.70$, $ps < 0.001$), and WM network strength was significantly more predictive of block-wise 2-back performance than any of the 8 canonical networks ($Zs > 3.33$, $ps < 0.001$). Both networks also outperformed 200 random size-match networks ($ps < 1/200$).

## Study 1.3. Working memory network strength predicts subsequent memory

Do youth with FC signatures of stronger sustained attention and/or working memory function during memory encoding show better later visual recognition memory? To investigate this question, we measured the relationship between recognition memory performance for $n$-back task stimuli to individual differences in networks strength values during the $n$-back task.

Recognition memory was assessed after scanning sessions. The $n$-back recognition memory test included 48 "old" stimuli (which has been presented during the $n$-back task) and 48 "new" stimuli (which has not been presented), with 12 images each of happy, fearful, and neutral faces as well as places. Participants were asked to rate each picture as either "old" or "new." Memory performance was measured as the discrimination index ($d'$) based on all stimuli. Recognition memory d' was related to strength in the SA and WM networks averaged across all blocks (i.e., both 0-back and 2-back blocks). Results revealed that strength in the WM ($r = 0.12$, $\rho = 0.13$, $p_{adj} < 0.001$), but not the SA ($r = 0.01$, $\rho = -0.01$, $p = 0.55$), network predicted subsequent recognition memory (**Fig C in S1 Text**). The strength of the WM network was also a significantly better predictor of subsequent memory than any of the 8 canonical functional networks (comparison of beta coefficients: $Zs > 5.37$, $ps < 0.001$) or random networks ($ps < 1/200$).

Unsurprisingly, in-scanner $n$-back performance was correlated with subsequent recognition memory performance across participants ($r = 0.31$, $p < 0.001$) [the correlation of recognition memory $d'$ with 0-back and 2-back accuracy separately is $r = 0.26$ and $r = 0.30$, respectively]. Nevertheless, the relationship between WM network strength and subsequent recognition memory remained significant even when in-scanner $n$-back performance accuracy was included in the regression model as a predictor ($\beta = 0.04$, t = 2.24, $p = 0.025$; **Table 3**) along with the age, sex, and residual head motion. Thus, the variance in recognition memory performance captured by the WM network is not fully accounted for by in-scanner task performance. This result highlights the unique contribution of the connectivity-based measures to long-term memory predictions.

**Table 3. WM network strength during in-scanner $n$-back task performance is related to subsequent recognition memory for $n$-back task stimuli after adjusting for nuisance variables and even $n$-back performance itself.**

| Predictors | Recognition memory d' | | | | |
|---|---|---|---|---|---|
| | Beta | CI | t | p | pr²% |
| (Intercept) | 0.73 | [0.67, 0.79] | 23.31 | <**0.001** | _ |
| Age | 0.06 | [0.03, 0.09] | 3.87 | <**0.001** | 0.93 |
| Female | −0.02 | [−0.08, 0.04] | −0.65 | 0.514 | 0.03 |
| Motion | 0.01 | [−0.02, 0.04] | 0.74 | 0.459 | 0.05 |
| $n$-back Accuracy | 0.18 | [0.14, 0.21] | 10.62 | <**0.001** | 6.83 |
| WM network strength | 0.04 | [0.00, 0.07] | 2.24 | **0.025** | 0.32 |
| SA network strength | −0.02 | [−0.05, 0.01] | −1.34 | 0.179 | 0.07 |
| **Random effects (site)** | | | | | |
| ICC | $0.02_{site}$ | | | | |
| Observations (subjects) | 1,489 | | | | |
| Marginal R²/conditional R² | 0.104/0.120 | | | | |

Marginal and conditional R² statistics estimate fixed-effects R² and total (i.e., fixed + random effects) R², respectively, based on [29]. Marginal semi-partial $r^2$ is calculated for each variable in the regression ($pr^2$%) and shown as percentage of total variance uniquely explained by the predictor using package partR2 in R.

SA, sustained attention; WM, working memory.

## Study 2 overview

In Study 2, we directly compared the adult and preadolescent brain networks that support SA and WM. In **Study 2.1**, we benchmarked the performance of the predefined adult network models in 2 ways to assess the effects of cross-dataset, cross-task, and cross-age generalization on models' predictive power. In **Study 2.2,** we asked how networks predicting SA and WM are differently configured in children and adults.

## Study 2.1. Benchmarking the predictive power of the adult sustained attention and working memory network models

Although the adult SA and WM network models successfully generalized to predict inter- and intra-individual differences in *n*-back task performance in the ABCD Study sample, effect sizes were modest. In Study 2.1, we benchmarked these effect sizes in 2 ways. First, we asked how close the predictive power of the adult SA and WM models came to a model of general cognitive ability trained in the ABCD Study sample itself. (We did not train separate ABCD-specific SA and WM network models because the ABCD Study task battery does not include an out-of-scanner SA measure.) Second, we asked how close the predictive power of the SA model in Study 1 came to a theoretical maximum for the 0-back and 2-back tasks by applying the same model to data from the high-quality adult HCP sample. (We could not fairly perform this analysis with the WM model because it was defined using HCP data.) Finally, in a post hoc analysis, we trained a new network predictor of SA in adults using 0-back accuracy and tested its generalizability to ABCD sample's 0-back performance (to maintain same SA task in adults and children).

## Building a development-specific connectome-based predictive model

To ask how close the predictive power of the adult SA and WM models came to that of a "youth-specific" network predictor of general cognitive ability trained in the ABCD sample itself, we defined a new connectome-based model—the **cognitive composite** network model —using leave-one-site-out cross-validation in the ABCD Study dataset (see Methods). The cognitive composite network model was defined to predict children's average performance on 5 out-of-scanner NIH Toolbox tasks (i.e., their "cognitive composite" score; see Methods) because NIH Toolbox data were collected outside the scanner and the cognitive composite score was similarly correlated with 0-back accuracy ($r = 0.31$, $\rho = 0.32$, $p < 0.001$) and 2-back accuracy ($r = 0.33$, $\rho = 0.38$, $p < 0.001$). Thus, it was fair to use the cognitive composite net-work model to benchmark the predictive power of both the SA networks and WM network models. (In other words, a model built to predict cognitive composite scores would not be biased at the outset to better predict 0-back or 2-back accuracy.)

Demonstrating its utility for this analysis, the cognitive composite model successfully pre-dicted cognitive composite scores in left-out ABCD Study sites ($r = 0.295$, $\rho = 0.27$, $p < 0.001$ across all sites; see **Figs D** and **E in** S1 **Text**). The youth cognitive composite network (averaged over all the site-wise models and binarized at a threshold of 0.5) included edges spanning wide-spread cortical and subcortical-cerebellar areas (**Fig 5**).

Cognitive composite network strength during 0-back task performance predicted 0-back accuracy ($r = 0.23$, $\rho = 0.23$, $p < 0.001$) and strength during 2-back task performance predicted 2-back accuracy ($r = 0.32$, $\rho = 0.33$, $p < 0.001$) in children from left-out sites (**Fig 5**; **Table A in** S1 **Text**). This youth cognitive composite model significantly outperformed the adult WM model for predicting 2-back accuracy ($\beta = 0.27$, t = 10.76 versus $\beta = 0.10$, t = 3.89, $p < 0.001$ from bootstrapping). Surprisingly, however, the adult SA model's prediction of 0-back

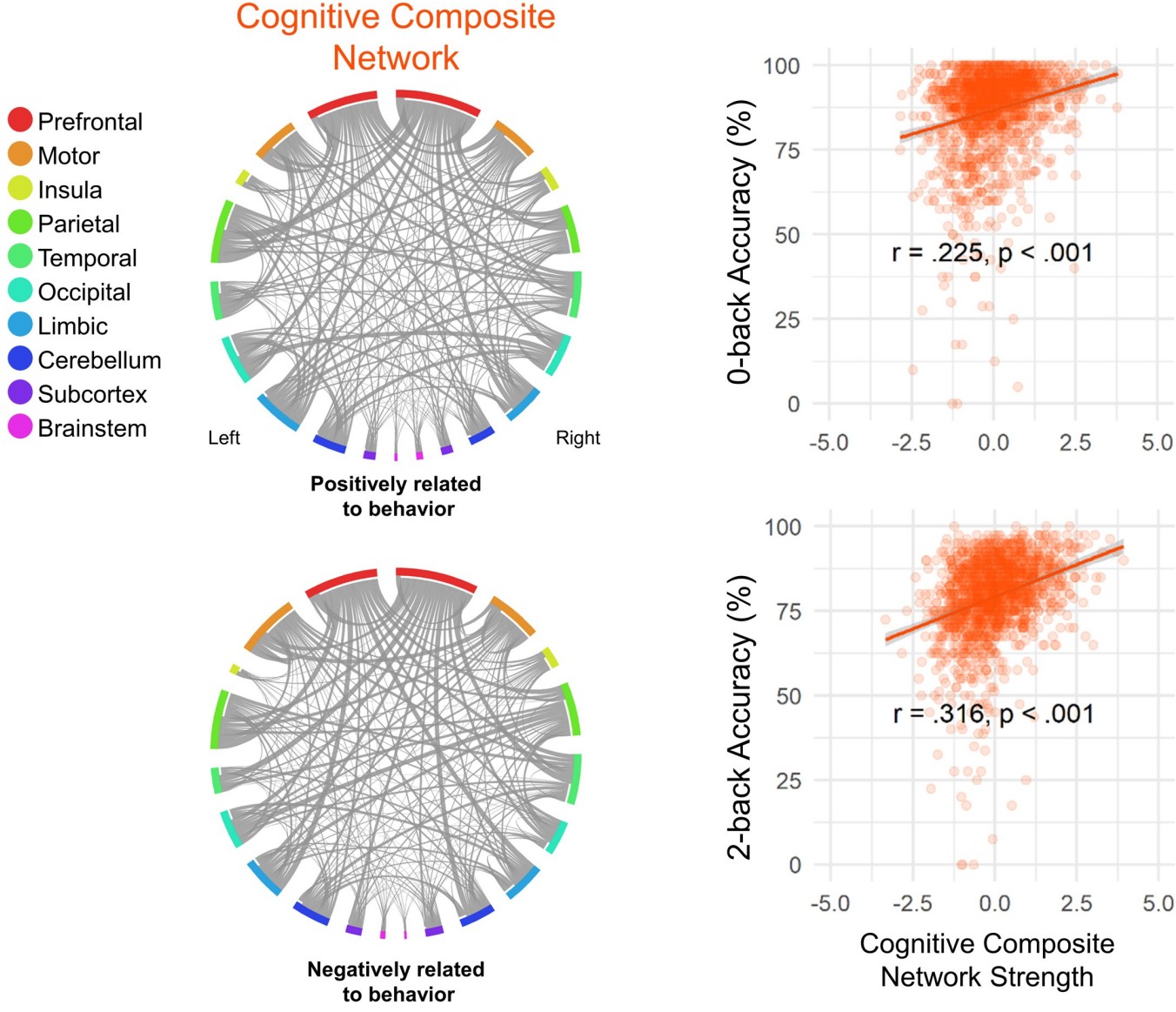

**Fig 5. Left:** The youth-defined cognitive composite network averaged over all the ABCD site iterations (binarized at a 0.5 threshold). **Right:** Cognitive composite network strength in 0-back and 2-back task blocks predict 0-back accuracy and 2-back accuracy across the ABCD sample, respectively. The data for this figure are available at NDA study 1849 10.15154/1528288. ABCD, Adolescent Brain Cognitive Development.

accuracy in youth was comparable with that of this ABCD-specific cognitive composite model ($\beta = 0.16$, t = 6.35 versus $\beta = 0.19$, t = 7.57, p = 0.242, N.S.). Furthermore, including both adult SA and youth cognitive composite network strength of youths in a regression model to predict their 0-back accuracy results in comparable $\beta$ coefficients for each ($\beta = 0.19$, t = 8.20 and $\beta = 0.21$, t = 7.59, respectively**; Table B in** S1 **Text**).

Finally, intra-individual differences analyses revealed that block-to-block changes in the strength of the youth cognitive composite network tracked block-by-block changes in both 0-back and 2-back accuracy. Echoing the block-by-block results observed with the adult network models in Study 1.2 (**Table 2**; SA network tracking 0-back accuracy: $\beta = 0.07$, t = 7.37; WM network tracking 2-back accuracy: $\beta = 0.04$, t = 3.83), the effects were significant but

subtle (**Table C in** S1 **Text**, cognitive composite network tracking 0-back accuracy: $\beta = 0.05$, t = 5.37; cognitive composite network tracking 2-back accuracy: $\beta = 0.08$, t = 8.33).

### Predicting *n*-back accuracy in adults

Compared to the original studies in which these networks were identified, both the SA [23] and WM [24] models show lower predictive power in the current study than they did in adults. This could arise for many reasons, including those related to developmental change (i.e., differences between adults and children) and unrelated to development (e.g., differences in scan sites and parameters and differences in the to-be-predicted behavioral task).

We used the HCP dataset to assess the degree to which differences unrelated to developmental change—scan site and parameters and task differences—impacted the predictive power of the SA model. To do so, we replicated the analyses in Studies 1.1 and 1.2 with *n*-back task HCP data and compared model performance to that achieved in the ABCD dataset. A result that the model predicted adults' 0-back accuracy better than it predicted children's would suggest that adult models do not capture well the functional networks underlying SA performance at age 9 to 11 and/or that predictive power was lower in the ABCD sample because of data quality. On the other hand, a result that the model did not predict adults' 0-back accuracy better than it predicted children's would suggest that adult models do capture the functional networks underlying sustained attention at age 9 to 11. In this case, predictive power may be lower in the ABCD Study sample than in adult datasets (e.g., [17]) because of site- or scanner-related differences or differences in the to-be-predicted behavioral measure of sustained attention (gradCPT *d'* in [23,17] versus 0-back accuracy in the ABCD and HCP samples).

HCP analyses included behavioral and fMRI data from 754 adults (405 female, 22 to 25 years old: 174, 26 to 30 years old: 321, 31 to 35 years old: 249, and 36+ years old: 10; see Methods). We applied the adult SA network mask to FC patterns of novel adults from HCP calculated during 0-back and 2-back blocks of the *n*-back task and related network strength to task performance both across and within subjects. (Again, we did not apply the adult WM network mask to HCP data because it was previously defined in this sample; [24]).

Demonstrating cross-dataset validity—and replicating the pattern of results observed in the ABCD sample—the adult SA network predicted individual differences in novel adults in 0-back accuracy ($r = 0.17$, $\rho = 0.12$, $p < 0.001$) but not 2-back accuracy ($r = 0.07$, $\rho = 0.07$, $p = 0.07$; **Fig F in** S1 **Text**), with the former correlation being significantly larger than the latter (Steiger's Z [test for the difference between 2 dependent correlations with different variables] = 2.29, $p = 0.02$). Results were consistent after adjusting for age, sex, and remaining head motion covariates (see **Table D in** S1 **Text**), and the $\beta$ coefficient was significantly larger for 0-back than 2-back accuracy ($\beta = 0.16$, t = 4.51 versus $\beta = 0.07$, t = 1.95; $p = 0.034$ from a bootstrap test). Mixed-effects regressions showed that, within-subject, block-by-block changes in SA network strength tracked block-by-block changes in 0-back accuracy ($\beta = 0.08$, t = 6.36, $p < 0.001$; **Table E in** S1 **Text**). Thus, the adult SA network generalized to a novel sample of adults to predict 0-back, but not 2-back, task performance, which is similar to what we had also observed in the ABCD sample (Study 1).

The predictive power of the adult SA network was numerically similar for children's and novel adults' 0-back task performance (between-subjects: ABCD $r = 0.19$ golden line in **Fig 6**, HCP $r = 0.17$ purple line in **Fig 6**). This was also true for tracking changes in performance within subjects (ABCD $\beta = 0.07$, t = 7.37; HCP $\beta = 0.08$, t = 6.36). This suggests that the SA network model captures children's 0-back (i.e., sustained attention) performance just as well as it captures adults'.

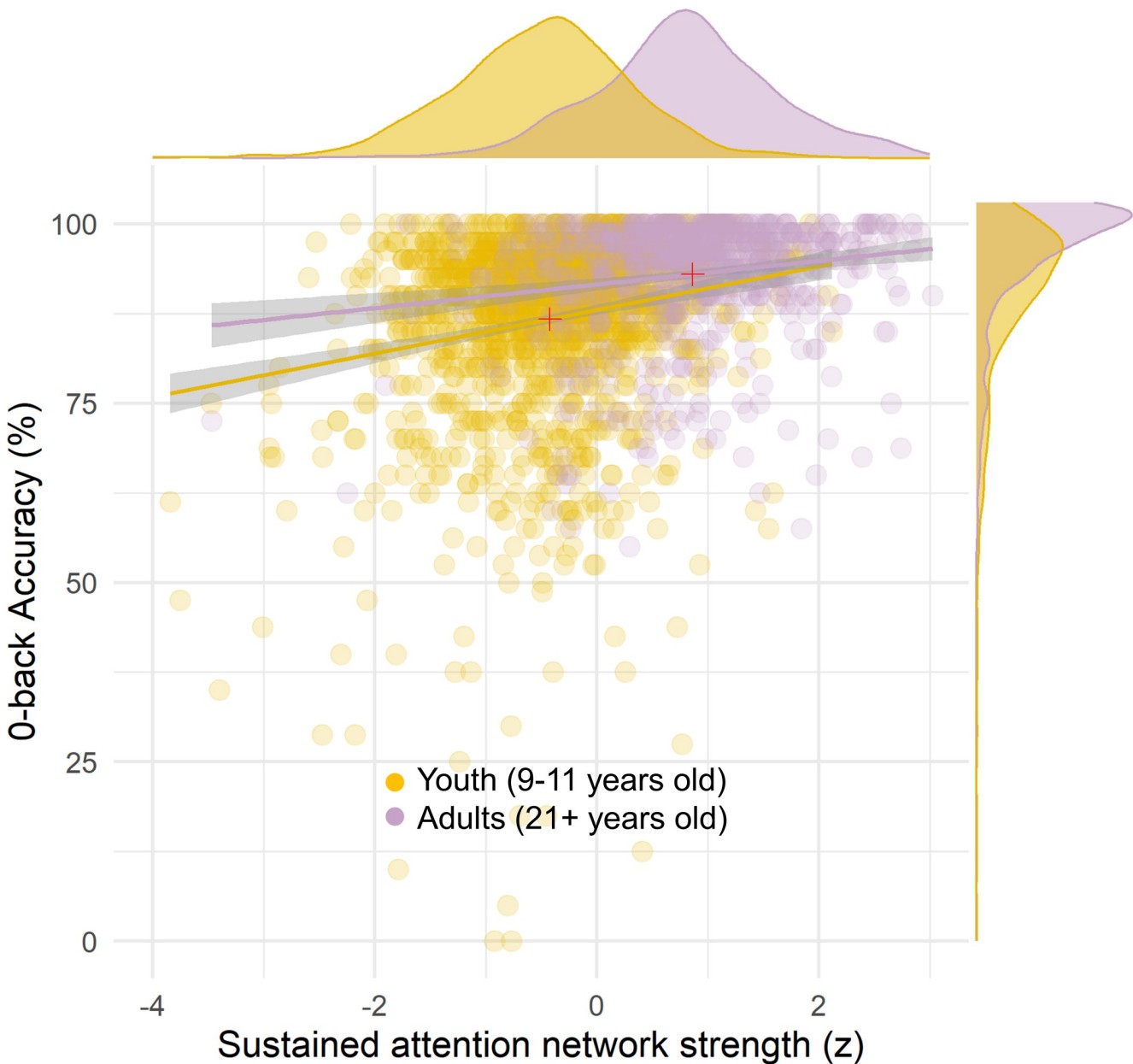

**Fig 6. The strength of the adult SA network predicts 0-back accuracy in youth and novel adults.** Even though the discriminability of individuals' task performance is not significantly different within each dataset, i.e., there is no significant difference between the correlations (ABCD = gold, $r = 0.19$, $\rho = 0.15$, ps < 0.001; HCP = violet, $r = 0.17$, $\rho = 0.12$, ps < 0.001; difference between $r$s is not significant $Z = 0.46$, $p = 0.456$), the mean performance and mean network strength are both significantly larger in adults. Overall, adults show stronger SA networks and better 0-back performance than youth (red + signs show the mean of SA strength and 0-back accuracy for the scatterplots in each dataset). The data for this figure are available at NDA study 1849 10.15154/1528288. ABCD, Adolescent Brain Cognitive Development; HCP, Human Connectome Project; SA, sustained attention.

Notably, the behavioral performance (mean 0-back accuracy) for adults is higher than youth by 6.2% ± 0.8% (**Fig 6**; Welch 2 sample t(2,130.6) = 14.3, $p < 0.001$), suggesting that SA ability is on average stronger in adults than youth. There are multiple possibilities that could explain this. One is that the preadolescent SA network is a "pre-mature" version of the adult SA network and predicts individual differences in youths similar to adults but is on average expressed less strongly at 9 to 11 years old. For example, in **Fig 6**, the SA network expressed in

the functional connectome of the adults is stronger than the 9- to 11-year-olds (Welch 2 sample t(1,448.9) = 35.8, $p < 0.001$). FC matrices were z-scored <u>within</u> each participant when used to quantify the SA network strength values for this analysis, but scanner differences between HCP and ABCD studies (e.g., spatially non-homogenous differences between HCP and ABCD scans unrelated to development) could still bias the group-average network strength values. Therefore, the current analyses cannot verify this explanation until longitudinal data from the same youths are processed, allowing mediation tests.

It is also possible that different subcomponents of the adult SA network predict task performance in adults and youth. To investigate this possibility, we "computationally lesioned" edges with at least 1 node in each of 10 macroscale brain regions from the SA model. We compared the effects of computational lesioning on the prediction of 0-back accuracy in the HCP and ABCD samples by comparing the $\Delta R^2$ in lesioned versus the full SA network strength models. We found that lesioning the prefrontal and temporal lobes decreased prediction power more in adults than it did in children ($p = 0.015$ for the prefrontal and $p = 0.007$ for the temporal lobe based on bootstrap distribution of $\Delta R^2$ values). Lesioning the subcortex, on the other hand, decreased prediction power more in children than it did in adults ($p = 0.009$). Therefore, the full SA network generalized equally well to adults and youth, though features within this network may contribute differentially to prediction at different ages.

Together, Studies 2.1.1 and 2.1.2 demonstrate that the adult SA model captures youth's individual differences and fluctuations in attention just as well as it captures novel adults'— and it is no worse at predicting attention in youth than is a youth-specific model of cognition defined in the ABCD dataset itself. Furthermore, the adult WM model captures general aspects of attention and memory in youth and is outperformed by a youth model of general cognition, potentially suggesting less consistency in the functional architecture of WM versus SA from age 9 to 11 to young adulthood.

## Alternative adult-defined connectivity-based predictor of sustained attention

To further assess the finding that there are more differences in the functional architecture of WM versus SA from preadolescence to adulthood, we trained a new connectivity-based predictive model for 0-back performance in adults from the HCP dataset's *n*-back fMRI data in a post hoc analysis. We then applied this HCP-based network predictor of SA instead of the Rosenberg and colleagues [23] SA model (which is trained on adults performing a gradual-onset CPT) to ABCD Study data. Consistent with our findings using the Rosenberg and colleagues [23] SA model, the HCP-based network predictor of 0-back in adults predicts 0-back accuracy in the youth sample ($r_{adj} = 0.25$, $p < 0.001$) but does not predict 2-back performance ($r_{adj} = 0.05$, $p = 0.056$). Importantly, we found that this alternative adult SA model fits the 0-back performance of the youths significantly better than the adult WM model predicts 2-back accuracy in youth ($r_{adj} = 0.11$; $p < 1/1,000$ when comparing fits based on bootstrapped distribution of $r_{adj}$ between the 2 models). This analysis further supports our finding that networks supporting SA are more consistent between youths and adults compared to those supporting WM.

## Study 2.2. Differences between adult and preadolescent networks

Next, we asked, in a data-driven manner, how the networks are differently configured in children and adults and how these differences relate to SA and WM performance. This analysis is distinct from the lesion analysis in Study 2.1.2 in that no predefined networks are being used to restrict the establishment of brain-behavior relationship differences between youth and adults.

To this end, we combined all whole-brain functional connectomes from the ABCD and HCP datasets. We then applied a partial least squares (PLS) regression analysis [31,32] that finds the linear combination of functional connections that maximally explains the age X 0-back performance covariance in these data (see Methods). This analysis revealed the multivariate patterns of FC that robustly covaried with latent variables (LVs) of age (child versus adult) and SA performance simultaneously. This process was then repeated for the 2-back performance to find the common versus age-specific functional connections that predict WM performance in adults and children.

Results of this analysis reveal 2 kinds of network configurations: one predicts cognitive performance across both age groups, while the other predicts performance specifically in youth or adults. In other words, PLS finds a pair of orthogonal LVs, one for brain-behavior relationships that are common for youths and adults and one for those that differ between youths and adults. Either the similarity or difference LV can emerge as dominant. None, one, or both LVs can be statistically significant with reliable loadings on connectivity and performance variables. Our expectation was that the commonalities would be greater than the differences based on the results of the previous studies showing comparable fit of the adult networks to the preadolescent brain. Critically, this analysis includes full connectomes and does not "constrain" functional networks to any predefined predictive models.

The SA PLS analysis primarily showed FC patterns that predicted better attentional performance in both ABCD Study youth and HCP adults (Fig 7, **top left**). In contrast, the second LV revealed the connections whose relationship to SA performance was dependent on the age group, showing connections that are robustly related to better SA performance in youths and worse performance for the adults (Fig 7, **bottom left**). Specifically, the primary LV consisting of 3,452 significant edges shows a pattern of connections that has a large positive correlation with 0-back accuracy in both youths ($r = 0.51$, CI = [0.43, 0.59]) and adults ($r = 0.63$, CI = [0.59, 0.67]). In contrast, the second LV consisting of 165 significant edges shows a pattern of connections that separate out the youths and adults in their task performance, as it is positively correlated ($r = 0.79$, CI = [0.74, 0.83]) with 0-back accuracy in preadolescents while negatively in adults ($r = -0.40$, CI = [-0.50, -0.28]).

A similar pattern of results was observed for the WM PLS analysis, where the first LV included functional connections that were related to WM performance similarly for youth and adults (Fig 7, **top right**), whereas the second LV showed functional connections that related to WM performance differentially for youth versus adults (Fig 7, **bottom right**). The primary WM LV consisted of 1,610 significant edges and predicted higher 2-back accuracy in youth ($r = 0.68$, CI = [0.62, 0.74]), as well as adults ($r = 0.66$, CI = [0.60, 0.71]). The second LV consists of 224 significant edges and is related to better 2-back performance in preadolescents ($r = 0.73$, CI = [0.66, 0.79]) but poorer performance in adults ($r = -0.63$, CI = [-0.72, -0.55]).

Importantly, these PLS results are consistent with our neuromarker generalizability approach by showing the first LV, which represents connections that are positively related to performance for both youths and adults, is stronger for SA than WM (cross-block covariance: attention $\sigma_{XY} = 0.692$; working memory $\sigma_{XY} = 0.624$, and these covariance scores are significantly different from one another $p < 1/200$ based on 200 bootstraps) [cross-block covariance $\sigma_{XY}$ is estimated as the ratio of the LV's squared eigenvalue over the sum of squared eigenvalues across all the LVs and represents the dominance of an LV in the PLS analysis]. Equivalently, LV 2 that represents functional connections that relate to performance differentially for youth versus adults is stronger for WM than SA (attention $\sigma_{XY} = 0.308$; working memory $\sigma_{XY} = 0.376$; $p < 1/200$ for the difference between these cross-block covariances). These analyses further support the idea of more differences between preadolescent and adult networks supporting working memory and more similarity between preadolescent and adult networks supporting sustained attention.

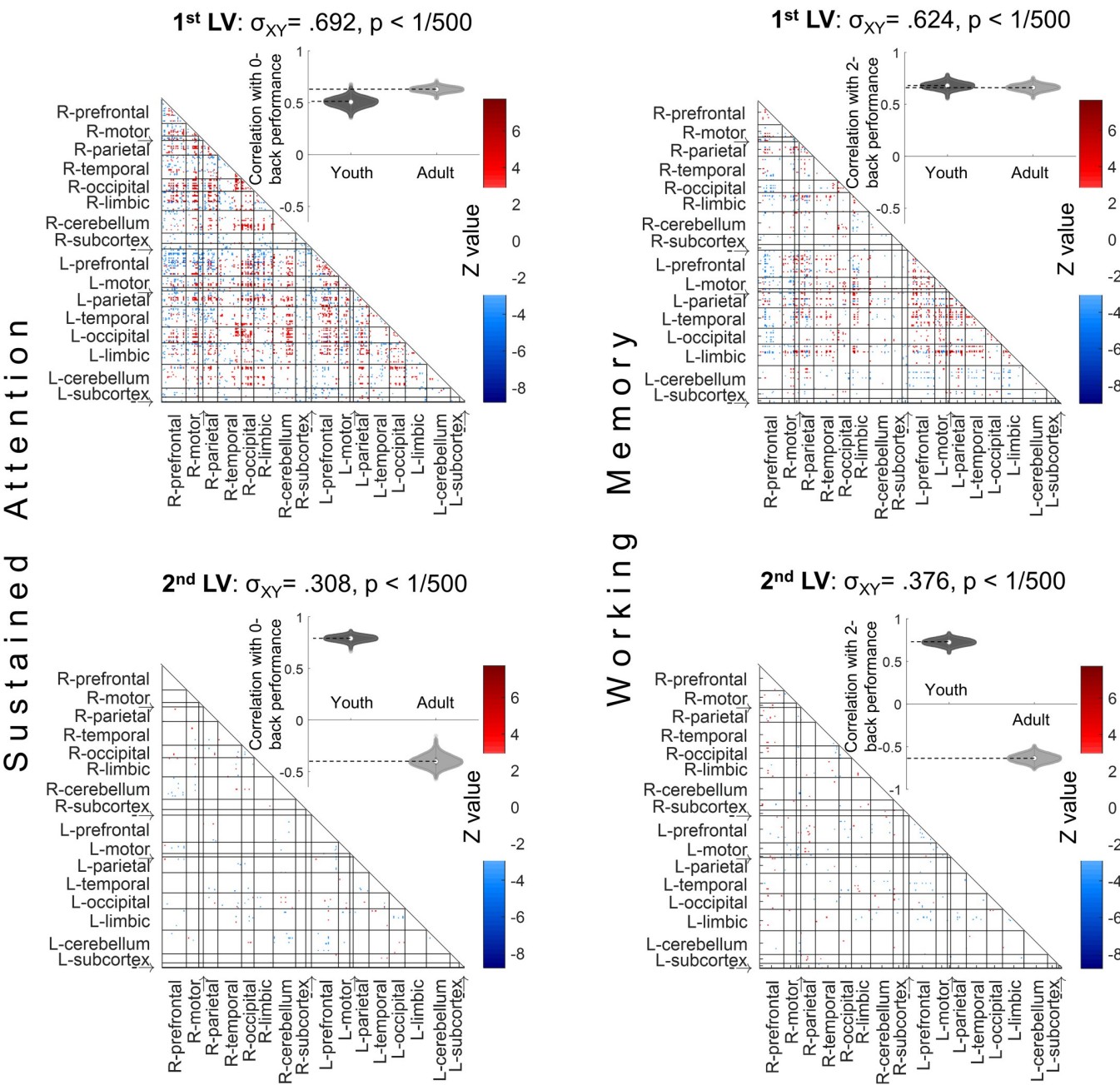

**Fig 7.** The data-driven LVs for the sustained attention (left column) and working memory (right column) PLS regressions. **Left column:** PLS relating functional connectivity to SA performance and age group successfully identified functional connectivity patterns related to better attentional performance in both ABCD youth and HCP adults (first LV), as well as connections differentially related to performance in youth and adults (second LV). **Right column:** PLS relating functional connectivity to WM performance and age group successfully identifies functional connectivity patterns related to better WM performance in both youth and adults (first LV), as well as connections differentially related to performance in youth and adults (second LV). *P* values are calculated from 500 permutations and LV weights are calculated from 500 bootstraps in each PLS; significant connections are those that have bootstrap ratios (Z) above +3 or below −3. We randomly selected 754 of the ABCD sample in each PLS to make the age group sizes balanced (HCP has *n* = 754) and the presented figure is averaged over 200 bootstrapped balanced samples. Using the full 1,545 ABCD participants instead of randomly balanced group sizes results in very similar PLS LVs. Anatomical labels: R and L refer to right and left hemispheres; "→" is insula and "-→" is brainstem. The data for this figure are available at NDA study 1849 10.15154/1528288. ABCD, Adolescent Brain Cognitive Development; HCP, Human Connectome Project; LV, latent variable; PLS, partial least squares; SA, sustained attention; WM, working memory.

## Discussion

Across 2 studies, we applied different approaches using connectome-based predictions to reveal differences in the functional architecture of SA and WM in preadolescence and adulthood. The first approach utilized adult network models developed previously (also a new connectome-based predictive model for 0-back accuracy) and allowed us to evaluate the degree to which these same networks predicted SA and WM in preadolescence. The second approach directly compared and contrasted the functional connectivity underlying SA and WM in youth compared to adults.

In Study 1, we found that connectome-based models of SA and WM previously defined in independent samples of adults generalized to capture inter- and intra-individual differences in SA and WM in 9- to 11-year-olds. In Study 2, we showed that the adult SA network predicted children's 0-back task performance just as well it predicted novel adults', and just as well as a youth-defined network predictor of general cognitive abilities. The adult WM model, on the other hand, predicted children's 0-back and 2-back (i.e., low- and high-working memory load) task performance, although not as well as the development-specific model of cognitive abilities. These results suggest that distinct functional brain networks predict SA and WM in the developing brain. Furthermore, WM network strength during the *n*-back predicted subsequent memory for the items in the recognition memory task performed later outside the scanner, even when adjusted for *n*-back performance. This result demonstrates that, in addition to predicting ongoing working task performance, WM network expression predicts future long-term memory.

The current work signifies 3 benefits of individualized predictive modeling with fMRI—and, in particular, of validating predictive markers in multiple independent datasets. First, training and testing brain-based predictive models allows us to investigate specific versus general brain markers of cognitive processes by conducting single- or double-dissociation analyses predicting individual differences in different aspects of cognition. This can inform the extent to which different processes relate to common or distinct functional network. For example, sustained attention and working memory are highly related processes as they covary together in individual ability [33,34], and attention lapses lead to worse working memory performance [35]. Additionally, the ability to control attention has been proposed to play a major role in complex WM tasks [36,37]. However, our results suggest that networks involved in SA are not sufficient to predict differences in WM (i.e., 2-back) performance across participants, despite predicting attentional (i.e., 0-back) performance in the same participants.

Second, we can ask whether the same networks that predict individual differences in behavior capture intra-individual change. Recent work has begun to suggest that fluctuations in large-scale functional brain networks index variance in SA function [17] and stimulus-unrelated thought [38] within individuals. In the current study, we demonstrate that block-by-block changes in SA network strength generalized to tracked block-to-block fluctuations in 0-back accuracy of children, and block-to-block fluctuations in WM network strength tracked block-by-block 2-back accuracy, above and beyond stimulus types and practice effects. This is remarkable given the relatively few volumes of data per block (30–31 TRs) and blocks per run. We also explored the source of changes in predictive network strength by investigating whether network fluctuations were more driven by (a) an individual's cognitive/attentional state fluctuations or (b) properties of the task stimuli that they saw (i.e., the category of *n*-back task images). We found that associations between network and *n*-back performance fluctuations are largely independent of the effects of stimulus type on performance. With further longitudinal data, it will be possible to directly model intra-individual changes in FC patterns that covary with performance, thus assessing the similarities and differences in trait-like versus state indicators of SA and WM processes in FC space.

Third, applying predictive models to predict differences in behavior both between and within individuals allows us to assess how brain systems underlying different cognitive processes differ (or remain consistent) across different time scales, from one moment, hour, and even year to the next. Although the adult neuromarkers of sustained attention and working memory both generalized to predict 9- to 11-year-olds' behavior, they differed in their comparative fit to youth versus novel adults. We also found that a new adult network predictor of sustained attention (based on 0-back performance in HCP) generalized to predict 0-back accuracy in the youths significantly better than the adult network predictor of working memory (based on 2-back performance in HCP) generalized to predict 2-back accuracy in youths. This may reflect differential developmental effects for each of the 2 cognitive constructs.

For WM, the adult network's predictive power in the ABCD sample was smaller than it was in the external validation sample of older adults in the Avery and colleagues [23] study (ABCD $r = 0.14$; older adults $r = 0.36$ [23]). Second, it was also smaller than the predictive power of the ABCD-defined cognitive composite network model (correlations with individual differences in 2-back accuracy: working memory network $r = 0.14$; cognitive composite network $r = 0.32$). These 2 comparisons converge to suggest the possibility of developmental differences in the functional architecture of WM in preadolescence and adulthood. Interestingly, there is a relatively large and significant overlap between the youth-defined cognitive composite network and the adult WM network both for edges positively related to behavior (11.6% overlap of combined edges, hypergeometric $p < 0.001$) and edges negatively related to behavior (10.7% overlap of combined edges, $p < 0.001$). This overlap may reflect a common subnetwork underlying general cognitive ability in youth and working memory in adulthood.

For SA, however, the cross-development results paint a different picture. The performance of the SA network in predicting 0-back performance was similar for youth and adults ($r = 0.19$ and 0.17, respectively). Although this may reflect similarity in the functional architecture of SA in these 2 age groups, it could arise from better "ground truth" prediction in adulthood disadvantaged by the relatively low variance in their 0-back accuracy (HCP SD = 0.081 versus ABCD SD = 0.126). Making this explanation unlikely, however, the child cognitive composite network did not significantly outperform the SA network in predicting 0-back accuracy ($r = 0.23$ versus 0.19, respectively) despite the identical variance in behavioral performance (i.e., both predictions are in ABCD sample). This makes the difference in variance between adults and children a less tenable explanation for similarities in prediction and instead indicates that the SA network is as informative about the functional architecture of sustained attention in youth as it is in adulthood. This is further corroborated by the good fit of the adult 0-back predictive model to the preadolescent 0-back performance ($r = 0.25$). The result suggests consistency in this architecture from preadolescence to adulthood. Additionally, the relationship between children's SA network strength and 0-back accuracy was unchanged when adjusting for the strength of the cognitive composite network, again pointing to its unique and specific relation to sustained attention even in youth.

Although the SA network model generalized equally well to novel youth and novel adult 0-back performance, the particular contributions of different anatomical regions involved in the network differed between the 2 populations. That is, lesioning prefrontal, temporal, and subcortical regions from the networks affected predictive power differently for youth and adults. This may be related to the fact that SA function does improve well into adulthood (Fortenbaugh and colleagues [7]), and in these samples 0-back performance was indeed higher in the HCP than the ABCD data (mean accuracy = 0.93 versus 0.87, $t$ (2,130.6) = 14.3, $p < 0.001$). Therefore, it may be the case that the neural markers of individual differences in SA are present and predictive by late childhood, but the way they are utilized to maintain focus on tasks may change through adolescence. Taking together these and the working memory results

discussed earlier, we found differential developmental effects for each of the 2 cognitive constructs, pointing to less consistency in the functional architecture of WM than that of SA from age 9 to 11 to young adulthood.

Study 2.2 echoed these results by demonstrating that the relationship between FC patterns and SA is more similar between youths and adults than the relationship between FC patterns and WM. Further, this analysis revealed that most functional connections related to behavioral performance (SA or WM) are shared between youths and adults, as the shared component was the primary LV for both SA and WM in the PLS regressions. However, there are also connections, which comprise the secondary LVs, that differentiate youths from adults, i.e., connections that predict better performance in youth but worse performance in adults. Future work can use longitudinal data to ask whether these patterns reflect neurodevelopment underlying cognitive and attentional performance improvement from preadolescence through adulthood.

When putting these results in the context of broader theories of functional brain development, our findings are consistent with 2 big picture frameworks. First, our finding that there is dynamicity in the brain-behavior mapping of both SA and WM in development (though not to the same extent for each process) is consistent with the interactive specialization hypothesis [39]. This framework posits that developmental change in cognitive skills or behavior is accompanied by widespread changes across network of regions, rather than maturation of single brain regions [39]. Second, we found that overlapping networks support SA and WM abilities between preadolescents and adults. This is consistent with the neural reuse hypothesis [40], which posits that regions of the brain are "reused" throughout development such that each will end up participating in multiple functions and involved in diverse tasks across multiple cognitive domains [41]. Prior empirical evidence supporting these 2 frameworks has been primarily from earlier stages of life than preadolescence and our approach, combined with other work in the development of brain structure (e.g., [42]) may shine some light on these theories in other stages of lifespan.

There are some limitations to this work. First, we rely on 0-back and 2-back performance to index SA and WM rather than more traditional tasks like a CPT and visual change detection tasks, as these more traditional paradigms are not included in the ABCD Study. Future work characterizing the generalizability of connectivity-based models to other tasks of attention and working memory can further inform their predictive boundaries. Additionally, work suggests that models can be generalizable and robust with features that vary (e.g., [43]). It is not the case that the edges in the SA and WM networks used here are the "end all be all" attention and working memory networks. Rather, they are a preregistered set of edges that robustly predict SA and WM function. Future work examining the overlap of edges that predict different measures of SA and WM in different datasets can help to refine the models to the minimum set of edges that still generalizes across datasets. We make progress towards this goal by making the 0-back HCP-based network predictor brain mask and the PLSR analysis brain mask available to other researchers. Second, the within-participant fluctuation effects in Study 1.2 are statistically significant yet modest, and the across-participant effects in Studies 1.1 and 1.3 have effect sizes that are small to medium. This is consistent with a recent report using the ABCD Study data [21], where, on average, out-of-sample multivariate brain-behavior associations (mean $r$ = 0.17) were smaller than in-sample associations (mean $r$ = 0.46). However, it is important to mention that these smaller effect sizes from large samples are reported to be more robust and replicable than larger effects sizes from small samples [21]. Although the current approach demonstrates the statistical significance and theoretical implications of conservative external model validation analyses, further work is needed to determine the practical significance and potential translational utility of these and other brain-based predictive models. Third, and as mentioned before, in Study 2.2, since the ABCD and HCP samples are not the same cohort, it

is difficult to tease apart the age-related versus scanning-related differences between the 2 samples in the multivariate PLS regression. However, it should be mentioned that the relative comparison of the primary LVs' dominance in SA versus WM PLS regressions would not be sensitive to the non-age-related factors unless these non-age-related factors interact differentially with fMRI data during 0-back versus 2-back blocks.

In conclusion, we found that distinct functional brain networks predict SA and WM abilities across youth, as well as changes in attentional and memory performances over time. Therefore, sustained attention and working memory are overlapping but distinguishable cognitive constructs in the preadolescent brain, with FC patterns of working memory differing more between youth and adults than those of sustained attention.

## Methods

### Data

We analyzed a subset of baseline-year behavioral and fMRI data from the ABCD (Release 2.0.1). The total sample in the dataset is 11,875 children 9 to 11 years old from 21 sites across the United States. We first excluded the participants who were scanned using Philips scanners (see fMRI data processing) or those without functional MRI data, resulting in 9,446 participants from 19 sites. After a visual quality check of all structural and functional scans, 4,939 of these participants had structural and at least 1 run of *n*-back task fMRI data that passed our visual quality check and had corresponding EPrime files containing trial-by-trial *n*-back task data. Next, we applied a frame displacement (FD) threshold of FD mean < 0.2 mm and FD max < 2 mm to remove *n*-back fMRI runs with excessive head motion, resulting in 1,839 participants. Finally, we removed *n*-back runs for which the start time of the behavioral recording file was unclear with respect to the fMRI data, or if the data were flagged for "switched box" or "*n*-back task done outside scanner", resulting in sample size of $N = 1,548$. Additionally, participants from 1 site with only $N = 3$ subjects after prior exclusions were not included in the across-subjects models that include site as a random intercept factor and also the within-subject analyses for consistency. Therefore, the final sample size was $N = 1,545$ participants 9 to 11 years old from 18 sites, mean age = 10.03 years old, 851 female.

### In-scanner emotional n-back task

The emotional *n*-back task in the ABCD dataset [44] includes 2 runs of 8 blocks each with 10 trials in each block. A picture is shown in every trial and participants are told to make a response on every trial, indicating whether the picture is a "Match" or "No Match." In each run, 4 blocks are 2-back task for which participants are instructed to respond "match" when the current stimulus is the same as the one shown 2 trials back. The other 4 blocks are of the 0-back task for which participants are instructed to respond "match" when the current stimulus is the same as the target image presented at the beginning of the block. At the start of each block, a 2.5 s cue indicates the task type ("2-back" or "target =" and a photo of the target stimulus; see **Fig G in** S1 **Text**). A 500 ms colored fixation precedes each block instruction, to alert the child of a switch in the task condition.

Two blocks of 0-back and 2 blocks of 2-back contain happy faces (1 in each run), another 2 in each task contain fearful faces, another 2 contain neutral faces, and another 2 contain places. There are 24 unique stimuli per type presented in separate blocks, each trial is 2.5 s (2 s presentation of a stimulus, followed immediately by a 500 ms fixation cross) resulting in 160 total trials in 16 blocks of *n*-back. Four fixation blocks (15 s each) also occur in each run after of every other *n*-back block.

## Post-scan *n*-back recognition memory task

We analyzed data from the post-scan *n*-back recognition memory task, in which 48 "old stimuli" (previously presented during the in-scanner emotional *n*-back task) and 48 new stimuli were presented to participants. Participants were asked to rate each picture as either "Old" or "New." Each picture was presented for 2 s followed immediately by a 1-s fixation cross. The 96 pictures shown have equal numbers of each stimulus type in the old and new stimulus sets (12 each of happy, fearful, and neutral facial expressions and places in each set).

## Out-of-scanner cognitive composite score

The "cognitive composite" behavioral scores were measured from each child's average performance in 5 out-of-scanner NIH-Toolbox tasks: the Picture Vocabulary task, Flanker inhibitory control and attention task, Pattern Comparison processing speed task, Picture sequence memory task, and Oral Reading recognition task. These tasks were chosen because they capture a wide range of cognitive processes and are the only NIH Toolbox tasks collected in subsequent ABCD data releases. This makes them (and the cognitive composite score used here) suitable for future longitudinal tracking of the general cognitive abilities of the children using the model developed from the current release. The mean of these 5 measures was used as the cognitive composite score rather than first principal component (PC) because the correlation between the mean and the first PC was $r = 0.94$. Therefore, we used the mean for a straightforward interpretation.

## Functional MRI data processing

Minimally preprocessed functional and structural scans for ABCD Release 2.0.1 were downloaded for all participants from the National Institutes of Mental Health data archive. Use of the data was approved by the relevant University of Chicago Institutional Review Board. Minimal preprocessing included motion correction, B0 distortion correction, gradient warping correction and resampling to an isotropic space [45]. Participants who were scanned on Philips brand scanners were excluded because of a known error in the phase encoding direction while converting from DICOM to NIFTI format. Next, a custom modification of the FMRI-PREP pipeline was run on all images. Each participant's structural T1w scan was skull-stripped, segmented by tissue type, and then normalized to the MNI152 nonlinear sixth generation template: the standard MNI template included with FSL. Functional scans were then aligned and normalized to the T1w space and then to MNI space and potential confounds of interest were extracted. Next 36 confounds [46] were regressed out of the voxelwise BOLD time series including: global mean signal, mean cerebro-spinal fluid signal, mean white matter signal, the 6 standard affine motion parameters and their derivatives, squares, and squared derivatives. This was followed by applying a bandpass filter with a high-pass cutoff of 0.008 Hz and a low-pass cutoff of 0.12 Hz via the 3dBandpass command in AFNI. Finally, the cleaned volumetric BOLD images were spatially averaged into 268 predefined parcels, including cortical, subcortical, and cerebellar regions, from the whole-brain Shen functional atlas [47].

## Predictive network anatomy

The **sustained attention** network mask (**Fig 2**) was defined to predict SA function using data collected from 25 adults who performed a gradual-onset continuous performance task (gradCPT; [48]) during fMRI [23]. Sustained attentional abilities were operationalized as participants' sensitivity (*d'*) on the gradCPT. The same functional networks that predicted performance in the initial training sample—a "high-attention" network whose strength predicted

higher $d'$ scores and a "low-attention" network whose strength predicted lower $d'$ scores—have generalized to independent datasets to predict performance on other attention tasks from data observed during rest and task performance. These include a stop-signal task (Rosenberg and colleagues [16]), Attention Network Task [49], Stroop task [50], and Sustained Attention to Response Task [51].

The SA networks do not rely on canonical brain networks, such as the default mode and frontoparietal networks, to predict sustained attention. Instead, the high- and low-attention networks comprise 757 and 630 functional connections, or edges, respectively (out of 35,778 total), and span distributed cortical, subcortical, and cerebellar regions. In general, functional connections between motor cortex, occipital lobes, and cerebellum predict better sustained attention whereas functional connections between temporal and parietal regions, within the temporal lobe, and within the cerebellum predict worse attention. Computationally lesioning the high- and low-attention networks by removing connections from specific brain networks does not significantly reduce predictive model performance [23], suggesting that the SA connectome-based predictive model does not rely on individual canonical networks to achieve significant prediction.

The **working memory** network mask (**Fig 2**; [24]) was defined to predict 2-back task accuracy from data observed during 10-min $n$-back task fMRI runs (both 0-back and 2-back blocks) in the HCP dataset ($N$ = 502 from the S900 release). Like the SA networks, the WM networks were defined using connectome-based predictive modeling [30,52]. Briefly, in this approach, a "high-working-memory" network whose strength predicted higher 2-back accuracy and a "low-working-memory" network whose strength predicted lower 2-back accuracy were identified by correlating all edges (defined with the 268-node whole-brain Shen atlas; [47]) with 2-back accuracy across the HCP sample and retaining the edges significantly related to performance ($p < 0.01$). The resulting network model predicted unseen 2-back accuracy scores in HCP sample ($r = 0.36$, $p < 0.001$) in an internal cross-validation analysis, and generalized to predict individual differences in a composite of visual and verbal memory task performance ($r = 0.37$, $p < 0.001$) from resting-state fMRI in an independent sample of 157 older adults, 109 of whom were memory-impaired [24].

The WM networks comprise a distributed set of edges (1,674 edges in the high-working-memory network and 1,203 edges in the low-working-memory network) including frontoparietal, subcortical-cerebellar, motor, and insular edges. Additionally, default mode network (DMN) connections are included in the WM networks, with the DMN and DMN-associated regions (including limbic, prefrontal, parietal, and temporal cortices) overrepresented in the high-working-memory network relative to the low-working-memory network [24].

### Functional connectivity measures

The stimulus onset and offset times of the first and last trial in each block of the ABCD $n$-back task data were extracted from each participant's $n$-back EPrime file (shared in the curated MRI data folders sourcedata/func/task_events). The node-wise BOLD signal time series during each $n$-back block (30 or 31 TRs; TR = 0.8 s; approximately 25 s from the onset of the first stimulus to the offset of the last stimulus in each block) were used to create block-wise functional connectivity matrices (FC matrices) by computing all pairwise Pearson correlations between the block-wise time series of the 268 Shen parcels. (See Defining $n$-back blocks with a temporal lag in the supplement for a replication analysis with different block onsets and offsets.) The positive edge mask (i.e., the functional connections positively related to behavior) and negative edge mask (i.e., the functional connections negatively related to behavior) for each of the SA and WM predefined networks (**Fig 2**) were then multiplied by the FC matrices. These network masks are 268*268 trinary matrices determining if a pairwise correlation

(edge) belongs to a certain predictive network with −1 or +1 or not with a 0. These are shown in Fig 2 and briefly described in the previous section. Next, the Fisher's z-transformed correlation values in the masked FC matrices were summed (with mask weight sign) to calculate the corresponding network's strength in the block for the participant:

$$\text{Block−wise network strength} = \sum_{i,j}^{268} tanh^{-1}(r_{i,j}) * w_{i,j}.$$

Where $r_{i,j}$ is the Pearson's correlation between BOLD time series of parcels $i$ and $j$, and $w_{i,j}$ is the corresponding network mask value of 0, 1, or –1.

The block-wise network strengths were averaged over all 0-back blocks or all 2-back blocks for Study 1's across-participant analyses and z-scored across participants. For Study 2, the measures were left at the level of blocks within each participant. In Study 4, the network strengths were averaged over all blocks (i.e., both 0-back and 2-back) for across-participant subsequent memory analysis, because the released recognition memory scores (d') were from stimuli that could have been encoded during 0-back and/or 2-back task blocks, and files distinguishing the subsequent memory stimuli source were not available for this ABCD data release.

## Youth cognitive composite network

For consistency, the youth cognitive composite model was constructed using connectome-based predictive modeling [52], the same approach used to define the adult SA and WM network models. To construct the cognitive composite network mask for ABCD participants from site $k$, we retained the FC matrices (calculated from the entire $n$-back task time series) and cognitive composite scores of participants from all sites excluding $k$. We correlated the strength of every FC with cognitive composite score across participants in this training set. The edges positively and negatively correlated with cognitive composite score ($p < 0.01$) defined the masks that were applied to the block-wise FC matrices of participants from the left-out site $k$ as described in the previous section. This analysis included 1,536 participants because 9 participants did not have all 5 NIH Toolbox measurements).

## HCP 0-back accuracy network

We constructed the connectome-based predictive model predicting 0-back accuracy in the HCP sample to be used on the ABCD sample using all 754 HCP participants. To be consistent with the Avery and colleagues [24], working memory CPM, fMRI data from all of the $n$-back task were used to construct the FC matrices. The edges positively and negatively correlated with 0-back accuracy ($p < 0.01$) defined the masks that were then applied to the block-wise FC matrices of participants from the ABCD Study sample. This 0-back predictor included 2,506 edges in the positive network (112 shared edges with the Rosenberg and colleagues SA positive network of 757 edges; hypergeometric $p < 0.001$, i.e., significant overlap) and 3,191 edges in the negative network (42 edges shared with the SA negative network of 630 edges; hypergeometric $p = 0.977$, i.e., no significant overlap). The network strength in this 0-back predictive network and the Rosenberg and colleagues [23] SA predictive network were correlated across the ABCD participants ($r_{adj} = 0.36$ during 0-back blocks and $r_{adj} = 0.18$ during 2-back blocks, $ps < 0.001$).

## Mediation analysis

To perform the mediation analysis described in **Fig A in** S1 **Text**, 2-back task performance and working memory network strength measures were mean-centered within each participant to remove individual differences and then entered into the mediation model. The model

included block type (Neutral Face versus Place) as the predictor, block-wise 2-back performance as the dependent variable, block-wise working memory network strength as the mediator, and block-wise FD during 2-back blocks as a covariate. The mediation was performed using the mediation package in R with the built-in bootstrapping option for computing $p$ values for the coefficients and the mediated effect.

### Hypergeometric cumulative distribution function

To assess whether the overlap of the edges of different predictive networks was statistically significant, we calculated the probability of the overlap being due to chance using the hypergeometric cumulative distribution function implemented in MATLAB (www.mathworks.com). The function used was *hygecdf()* computed as:

$$F = \sum_{i=0}^{x} \frac{\binom{K}{i}\binom{M-K}{N-i}}{\binom{M}{N}}.$$

Where F is the probability of drawing up to x of a possible K items in N drawings without replacement from a group of M objects. The $p$ value for significance of overlap is then calculated as 1-F.

### Removing relatives does not change across-participant results

In our final sample ($n = 1,545$), there were 82 related children (41 pairs). We repeated the across-participant analyses after randomly removing from our sample 1 sibling from each pair (new $n = 1,504$) and found no significant differences in the results (see **Tables F–H in** S1 **Text**). We did not repeat the block-to-block change analyses because within-participant analyses are not affected by across-participant relationships.

### Non-participation and post-stratification weights do not change across-participant results

Participant exclusion rates are high for the functional neuroimaging data in adolescents, mainly due to high attrition rate from head motion, including the current study. To correct for any biases due to not being included in the analysis relative to the demographic characteristics of the overall ABCD Study sample, we conducted sensitivity analyses to confirm if the relationships in our between-subject analyses ($n = 1,545$) hold with non-participation (non-inclusion) weights (overall ABCD sample $n = 11,875$). Specifically, results were re-assessed with membership in the analyses weighted by estimated non-participation weights calculated for sex, age, race/ethnicity categories, Child Behavior Checklist (CBCL) mean score (square root transformed), and family income and then combined with the American Census Study (ACS) to ABCD raked propensity scores in file *acspsw03.txt* of Curated Release 2.0.1 (see [53,54]. The process is similar to [55,56]. In brief, inclusion/exclusion was regressed on age, sex, race/ethnicity, family income, and parental education in an elastic net regularized binary logistic regression model using glmnet in R. The logistic regression model picked the optimal tuning parameter lambda with the least cross-validation deviance in model selection. Having selected the optimal model, the non-participation weights are the inverses of the probabilities of response, conditional on being sampled. In order to compute corrected correlations between network strengths and task performances, non-participation weights capturing which individuals had available

data were multiplied by the post-stratification weights. Finally, the weighted mean, weighted variance, and weighted covariance were computed as in [57], to construct the corrected correlations. No between-subject results were changed with the non-participation and post-stratification weights. Within-subject analyses are not impacted by non-participation.

### Results are consistent with less motion restricted sample

Our main analysis uses a relatively stringent head motion criterion consistent with motion-related reliability issues in the task fMRI in youth (see [58]). However, in addition to the participation weight analysis, we also re-analyzed the ABCD data with a less stringent motion threshold of FD mean < 0.5 mm and FD max < 5 mm to retain more of the ABCD Study sample. The sample size for this FD threshold was $n$ = 3,225, which is double the sample with the more stringent FD threshold in the main analyses (FD mean < 0.2 mm and FD max < 2 mm; $n$ = 1,545). The results are shown in Tables J–M and Fig I in S1 Text and echo the same findings from the more restricted sample.

### Human Connectome Project data

In Study 2, we analyzed data from the HCP release S1200, a multisite consortium that collected MRI, behavioral, and demographic data from 1,113 participants. Minimally preprocessed, open-access $n$-back fMRI data were downloaded from connectomeDB (https://db.humanconnectome.org/) via Amazon Web Services. The acquisition parameters and prepossessing of these data have been described in detail elsewhere [59]. Briefly, preprocessing for task data included gradient nonlinearity distortion correction, fieldmap distortion correction, realignment, and transformation to a standard space. In addition, we applied additional preprocessing steps to the minimally preprocessed task data. This included a high-pass filter of 0.001 Hz via fslmaths [60] and the application of the ICA-FIX denoising procedure using the HCPpipelines (https://github.com/Washington-University/HCPpipelines) tool, which regresses out nuisance noise components effectively, similar to regressing out motion parameters and tissue type regressors [61]. The cleaned volumetric BOLD images were spatially averaged into 268 predefined parcels [47].

A total of 32 participants without sync time information files or motion regressor files for both $n$-back runs were removed from further analysis. Next, similar to the ABCD dataset, we applied a FD threshold of FD mean < 0.2 mm and FD max < 2 mm to remove $n$-back fMRI runs with excessive head motion, resulting in 881 participants. Finally, we removed participants with any quality control flags from the HCP quality control process (variable QC_Issue), resulting in a final sample of 754 participants.

FC measures for HCP data were computed as described for the ABCD data. Text files containing the timing information of the $n$-back trials were used to extract the beginning and ending of the blocks for each participant (each block approximately 35 TRs; TR = 0.72 s; approximately 25 s). An FC matrix for each block was constructed from the Pearson correlation between the BOLD signal time series of pairs of Shen parcels, and the sustained attention mask was applied to each block-specific FC matrix. The block-wise SA network strength values were averaged over all 2-back blocks or all 0-back blocks for Study 2.1's across-participant analyses and $z$-scored across participants. For Study 2.1's within-participant analysis, the measures were left at the level of blocks.

### Partial least squares regression

In Study 2.2, we used PLS to identify the relationship between the set of connections with group-by-performance accuracy. The PLS implementation software was downloaded from

Randy McIntosh's lab at: https://www.rotman-baycrest.on.ca/index.php?section=84. In PLS, the goal of the analysis is to find weighted patterns of the original variables in the 2 sets (termed "latent variables" or "LVs") that maximally co-vary with one another ([31,32]; examples of application for fMRI connectivity studies: [62–65]). Briefly, PLS is computed via singular value decomposition (SVD). The covariance between the 2 datasets X (FC matrices) and Y (age-group stacked task performances) is computed (X'Y) and is subjected to the SVD:

$$SVD(X'Y) = USV\prime.$$

Where U and V (the right and left singular vectors) provide weights (or "saliences") for the 2 sets (connectivity matrices and group-by-performance), respectively. The scalar singular value on the diagonal matrix S is proportional to the "cross-block covariance" between X and Y captured by the LV and is naturally interpreted as the effect size of this statistical association (reported as $\sigma_{XY}$).

First 200 re-samples of 754 participants from the 1,545 total ABCD sample were randomly selected and combined with the 754 HCP participants to make the age groups balanced. For each of these $2 \times 754$ samples, a set of 500 bootstrap samples were created by re-sampling subjects with replacement (preserving age labels) in order to determine the reliability with which each connection contributes to the overall multivariate pattern. Each new covariance matrix was subjected to SVD as before, and the singular vector weights from the resampled data were used to build a sampling distribution of the saliences from the original dataset. Saliences that are highly dependent on which participants are included in the analysis will have wide distributions, therefore low reliability. For the functional connections, a single index of reliability (termed "bootstrap" ratio or "$Z_{BR}$") was calculated by taking the ratio of the salience to its bootstrap estimated standard error. A $Z_{BR}$ for a given connection is large when the connection has a large salience (i.e., makes a strong contribution to the LV) and when the bootstrap estimated standard error is small (i.e., the salience is stable across many resamplings). Here, connections with $Z_{BR} > 3$ or $Z_{BR} < -3$ were selected as showing reliable increase or decrease in FC, respectively (equivalent to p~0.0025, 2-tailed, under normal distribution assumptions) similar to [66]. In each iteration, a set of 500 covariance matrices were generated by randomly permuting condition labels for the X variables (brain set). These covariance matrices embody the null hypothesis that there is no relationship between X and Y variables. They were subjected to SVD resulting in a null distribution of singular values. The significance of the original LVs was assessed with respect to this null distribution. The *p* value was estimated as the proportion of the permuted singular values that exceed the original singular value.

## Supporting information

**S1 Text. Appendix: Supplementary Analyses and Material. Fig A in S1 Text.** Task accuracy (top) and network strength (bottom) as a function of stimulus types. **Fig B in S1 Text.** Results of the mediation analyses relating block-by-block performance in the 2-back task to stimulus type, with fluctuations in working memory network strength as a mediator. **Fig C in S1 Text.** Working memory network strength, but not sustained attention network strength, was predictive of children's subsequent memory task performance. **Fig D in S1 Text.** Relationship between cognitive composite network strength and cognitive composite scores in the ABCD sample in all iterations of leave-one-site out combined. **Fig E in S1 Text.** Relationship between cognitive composite network strength and cognitive composite scores in the ABCD sample in each site separately is shown with blue points on permutation null distributions (violins). **Table A in S1 Text.** Model predicting individual differences in 0-back and 2-back task accuracy from cognitive composite network strength in the ABCD Study sample. **Table B in S1**

**Text.** Model predicting individual differences in 0-back and 2-back task accuracy from cognitive composite and sustained attention network strength in the ABCD Study sample. **Table C in S1 Text.** Model predicting intra-individual differences in 0-back and 2-back task accuracy from youth cognitive composite network strength in the ABCD Study sample. **Fig F in S1 Text.** Relationship between sustained attention network strength and $n$-back accuracy in the adult HCP sample. **Table D in S1 Text.** Model predicting individual differences in 0-back and 2-back task accuracy from sustained attention network strength in the HCP sample. **Table E in S1 Text.** Model predicting intra-individual differences in 0-back and 2-back task accuracy from sustained attention network strength and block type in the HCP sample. **Fig G in S1 Text.** The instruction and first 3 trials in a block of 0-back task (top, example from Places block type) and 2-back task (bottom, example from Positive Faces block type) are shown in this figure. **Table F in S1 Text.** Model predicting inter-individual differences in 0-back and 2-back task accuracy from sustained attention and working memory network strength values in the ABCD sample with only one of each pair of family members retained randomly ($n$ = 1,504). **Table G in S1 Text.** Model predicting inter-individual differences in 0-back and 2-back task accuracy from sustained attention and youth cognitive composite network strengths in the ABCD sample with only one of each pair of family members retained randomly ($n$ = 1,504). **Table H in S1 Text.** Model predicting out-of-scanner item memory recognition from sustained attention and working memory network strengths in the ABCD sample with only one of each pair of family members retained randomly. **Fig H in S1 Text.** The time-course of $n$-back blocks (from the beginning of the first trial's stimulus onset to the end of the last trial's stimulus offset time in the block) in a run in the ABCD n-back task. **Table I in S1 Text.** Working memory network strength measured in the 6-sec shifted manner described above is not significantly related to subsequent recognition memory for $n$-back task stimuli after adjusting for nuisance variables and $n$-back performance itself. **Table J in S1 Text.** Regression of n-back accuracy against network strength values across ABCD participants with liberal head motion criteria. **Table K in S1 Text.** Regression of n-back accuracy against network strength values in blocks within participants using ABCD participants with liberal head motion criteria. **Table L in S1 Text.** Regression of recognition memory performance against network strength values across ABCD participants with liberal head motion criteria. **Table M in S1 Text.** Regression of n-back accuracy against cognitive composite network strength values across ABCD participants with liberal head motion criteria. **Fig I in S1 Text.** PLS regression results including ABCD participants with liberal head motion criteria. (DOCX)

## Acknowledgments

Resources were provided by the University of Chicago Research Computing Center.

ABCD Acknowledgement: Data used in the preparation of this article were obtained from the Adolescent Brain Cognitive Development (ABCD) Study (abcdstudy.org), held in the NIMH Data Archive (NDA). This is a multisite, longitudinal study designed to recruit more than 10,000 children age 9 to 10 and follow them over 10 years into early adulthood. The ABCD Study is supported by the National Institutes of Health and additional federal partners under award numbers U01DA041022, U01DA041028, U01DA041048, U01DA041089, U01DA041106, U01DA041117, U01DA041120, U01DA041134, U01DA041148, U01DA041156, U01DA041174, U24DA041123, and U24DA041147. A full list of supporters is available at abcdstudy.org/nih-collaborators. A listing of participating sites and a complete listing of the study investigators can be found at abcdstudy.org/principal-investigators.html. ABCD consortium investigators designed and implemented the study and/or provided data

but did not necessarily participate in analysis or writing of this report. This manuscript reflects the views of the authors and may not reflect the opinions or views of the NIH or ABCD consortium investigators.

The ABCD data repository grows and changes over time. The ABCD data used in this report came from NIMH Data Archive Digital Object Identifier 10.15154/1504041. DOIs can be found at nda.nih.gov/study.html?id=721.

This research also benefited from the ABCD Workshop on Brain Development and Mental Health, supported by the National Institute of Mental Health of the National Institutes of Health under Award Number R25MH120869 and by UG3-DA045251 from the National Institute of Drug Abuse.

Secondary analysis of ABCD and HCP data was approved by the relevant University of Chicago Institutional Review Board.

## Author Contributions

**Conceptualization:** Omid Kardan, Monica D. Rosenberg.

**Data curation:** Omid Kardan, Andrew J. Stier, Carlos Cardenas-Iniguez, Kathryn E. Schertz, Julia C. Pruin, Taylor Chamberlain, Wesley J. Meredith, Xihan Zhang, Jillian E. Bowman, Tanvi Lakhtakia, Lucy Tindel.

**Formal analysis:** Omid Kardan, Andrew J. Stier.

**Funding acquisition:** Marvin M. Chun, Marc G. Berman, Monica D. Rosenberg.

**Investigation:** Omid Kardan, Emily W. Avery, Marc G. Berman, Monica D. Rosenberg.

**Methodology:** Omid Kardan, Andrew J. Stier, Carlos Cardenas-Iniguez, Yuting Deng, Monica D. Rosenberg.

**Project administration:** Monica D. Rosenberg.

**Resources:** Monica D. Rosenberg.

**Software:** Andrew J. Stier, Yuting Deng.

**Supervision:** Marvin M. Chun, Marc G. Berman, Monica D. Rosenberg.

**Validation:** Omid Kardan.

**Visualization:** Omid Kardan, Kathryn E. Schertz.

**Writing – original draft:** Omid Kardan.

**Writing – review & editing:** Omid Kardan, Andrew J. Stier, Carlos Cardenas-Iniguez, Kathryn E. Schertz, Julia C. Pruin, Taylor Chamberlain, Wesley J. Meredith, Xihan Zhang, Jillian E. Bowman, Tanvi Lakhtakia, Lucy Tindel, Emily W. Avery, Qi Lin, Kwangsun Yoo, Marvin M. Chun, Marc G. Berman, Monica D. Rosenberg.

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
