## [Editor Report · Decision Letter 0]

8 May 2022

Dear Dr Kardan, 

Thank you for submitting your manuscript entitled "Connectome-based predictions reveal developmental change in the functional architecture of sustained attention and working memory" for consideration as a Research Article by PLOS Biology.

Your manuscript has now been evaluated by the PLOS Biology editorial staff as well as by an academic editor with relevant expertise and I am writing to let you know that we would like to send your submission out for external peer review.

However, before we can send your manuscript to reviewers, we ask that you address two points that our academic editor has raised. First, as this is a cross-sectional design, they ask that you refer to developmental "differences" rather than "changes". Second, (and a larger request) the academic editor expressed some concerns with your analytical approach, suggesting that it would be stronger to use a model comparison to show that the selected model for one group/task is the best one compared to other potential models, rather than just successfully explaining the data. Once you've addressed these issues, please complete your submission by providing the metadata that is required for full assessment. 

To this end, please login to Editorial Manager where you will find the paper in the 'Submissions Needing Revisions' folder on your homepage. Please click 'Revise Submission' from the Action Links and complete all additional questions in the submission questionnaire. 

Once your full submission is complete, your paper will undergo a series of checks in preparation for peer review. Once your manuscript has passed the checks it will be sent out for review. To provide the metadata for your submission, please Login to Editorial Manager (https://www.editorialmanager.com/pbiology). Please note that we normally expect a fairly quick turn-around for this resubmission. However, given the alternative analysis approach requested by the academic editor, I've adjusted your "due date" for two weeks from now (May 24th). If you feel you'd need more time than that, please let me know.

If your manuscript has been previously reviewed at another journal, PLOS Biology is willing to work with those reviews in order to avoid re-starting the process. Submission of the previous reviews is entirely optional and our ability to use them effectively will depend on the willingness of the previous journal to confirm the content of the reports and share the reviewer identities. Please note that we reserve the right to invite additional reviewers if we consider that additional/independent reviewers are needed, although we aim to avoid this as far as possible. In our experience, working with previous reviews does save time. 

If you would like to send previous reviewer reports to us, please email me at kdickson@plos.org to let me know, including the name of the previous journal and the manuscript ID the study was given, as well as attaching a point-by-point response to reviewers that details how you have or plan to address the reviewers' concerns. 

Kind regards,

Kris

Kris Dickson, Ph.D.

Neurosciences Senior Editor/Section Manager

PLOS Biology

kdickson@plos.org

---

## [Decision Letter · Decision Letter 1]

6 Jul 2022

Dear Dr Kardan,

Thank you for your patience while your manuscript "Connectome-based predictions reveal developmental differences in the functional architecture of sustained attention and working memory" was peer-reviewed at PLOS Biology. It has now been evaluated by the PLOS Biology editors, an Academic Editor with relevant expertise, and by two new reviewers. We also took into account the prior peer review comments and your responses to those reviews.

In light of the prior feedback and the new reviews, which you will find at the end of this email, we would like to invite you to revise the work to thoroughly address the reviewers' reports and the additional feedback from our Academic Editor (also detailed at the bottom of this email).

Given the extent of revision needed, we cannot make a decision about publication until we have seen the revised manuscript and your response to the reviewers' comments. Your revised manuscript is likely to be sent for further evaluation by all or a subset of the reviewers.

**IMPORTANT - SUBMITTING YOUR REVISION**

*Re-submission Checklist*

*Published Peer Review*

*PLOS Data Policy*

*Blot and Gel Data Policy*

Sincerely,

Kris

Kris Dickson, Ph.D. (she/her)

Neurosciences Senior Editor/Section Manager

PLOS Biology

kdickson@plos.org

ACADEMIC EDITOR COMMENTS:

I would like the authors to consider the following additional changes/additions:

1) avoid discussing the word "change" in the context of developmental differences.

2) Having a supplementary analysis that uses less strict exclusion criteria due to head movements to examine some of the concerns by Reviewer 1. I think that nearly 2/3 exclusion of the sample at this age (9-11) is very rare, and I wonder why is it so high.

3) Discuss the implications of their findings for cognitive/biological theories of brain development such as Mark H. Johnson's interactive specialisation approach, and Anderson's neural re-use hypothesis.

REVIEWS:

Reviewer's Responses to Questions

PLOS authors have the option to publish the peer review history of their article (what does this mean?). If published, this will include your full peer review and any attached files.

Reviewer #1: No

Reviewer #2: Yes: Joe Bathelt

Reviewer #1: This very rigorous manuscript demonstrates that a priori connectomic brain measures used to predict sustained attention and working memory, in adult datasets, generalize to the predict a proxy for sustained attention (0-back performance) and working memory (the same 2-back task) in a group of children from the ABCD dataset. A main strength of this study is that it demonstrates that some aspects of functional brain architecture that support attention and WM in adults are present in children. The second main strength is that it further corroborates the utility and generalizability of these CPMs across datasets and different populations. On the other hand, there are weaknesses in some unwarranted conclusions drawn from these analyses that are overstated or misleading. 

1. From the title, and throughout the manuscript, differences in prediction accuracy across CPMs and datasets are used as evidence for "developmental differences" in sustained attention vs. WM. I do not think these conclusions are warranted, as I do not believe that differences in prediction of CPMs themselves drawn from different datasets (composite, SA, WM) can be easily interpreted. Similarly, ABCD vs. HCP datasets have many differences besides age, including scan parameters, data size, data quality, etc., thus prediction comparisons across datasets are difficult to interpret. To compound it, these statistically significant differences in prediction are notably small in terms of variance explained. Further, the SA CPM was based on a different task (not 0-back) and perhaps it captures similar variance in kids and adults because it doesn't generalize compared to training and testing on the same exact task. In other words, the SA CPM may be non-superior in adults because it is from a different task. Therefore, I do not believe the results support differential development of neural architecture for SA vs. WM.

2. While the generalization of these CPMs is exciting, the interpretation of the anatomy is potentially overstated. In the group's prior PNAS work, edge-overlap between models was modest. As such, while these edges are sufficient to predict SA and WM, they may not be necessary. A more rigorous lesion approach would remove these edges, and determine if an alternative CPM with orthogonal edges could predict. This would help justify anatomical specificity. In a related point, more traditional functional network-based parcellations (rather than these more strictly anatomical "networks") would help strengthen the argument that network-based prediction failed to adequately capture behavior, and this could be included in a supplement. Along these lines, if the composite CPM can predict WM and SA, then it further suggests that many different sets of connections can predict cognition broadly, and there is not specificity to the WM or SA CPMs per se. How well the weights/edges reproduce is key to interpretation of the anatomical specificity of CPMs.

3. Given these first two limitations, while the results do support some common neural architecture for SA and WM in children and adults, I don't believe this is a novel premise, as children activate overlapping regions as adults during cognition, and have at least similar functional network properties. Of course, I still applaud the novelty of the current study's methods, scope, and rigor.

4. Stepping back, even this conclusion that children use similar brain connections as adults has significant limitations. The sizable data exclusion necessary (>2/3) raises concerns about the generalizability to "normal" or most children for that matter. Excluding the large majority of the ABCD sample for too much movement or task performance leads to the real possibility that this study may be about the highest functioning children, and the most "adult-like" in the sample. This limits the study's generalizability to average children, as well as potentially the clinical utility of these findings (e.g., to ADHD). 

Other comments:

1. Given skewed performance distribution particularly in the 0-back, the authors should consider reporting Spearman rho or other non-parametric tests, especially given the relatively small parametric effect sizes.

2. Why not consider a 0-back CPM derived from the HCP data to compare more directly across the same tasks?

3. It would be interesting to consider generalizability across block types when predicting individual differences in study 1.1.

Reviewer #2: This paper reports the results of a data-driven analysis of functional connectivity in the ABCD and HCP datasets. There are many interesting results. Arguably, the main result is that connectome-based predictive models of sustained attention and working memory predict performance in children. However, a youth-specific model of working memory performed better compared to the adult model indicating more developmental differences in the underlying neural correlates. The analyses are based on a well-documented and replicated approach (connectome-based predictive modelling), use large and representative samples, and utilise cross-validation across datasets. The additional control analyses are extensive and very convincing. Further, the dicscussion sticks closely to the results and provides a well-rounded interpretation. I was only invited as a reviewer after the first round of revisions. I agree with the highly positive assessment of both reviewers regarding the sophisticated methodological approach. I do not agree with the first reviewer's assessment that the study does not add anything to the understanding of neurocognitive development. I think that the authors make a substantial contribution by showing that the functional connectivity correlates of sustained attention and working memory show different developmental patterns, especially because they use a data-driven and unconstrained methodological approach. I think that this study is an absolute triumph and will signify a new standard for conducting research within developmental cognitive neuroscience. In my view, the manuscript can be published without further revisions.

(Very) Minor comments:

Figure 6: Please explain what the arrow indicates, either in the caption or in the figure. 

Figure 7: The difference in correlation with 0-back performance between the youth and adult sample is very difficult to read. Please consider adding some horizontal lines, e.g. at .5 and .75

---

## [Editor Report · Decision Letter 2]

9 Nov 2022

Dear Dr Kardan,

Thank you for your patience while we considered your revised manuscript "Connectome-based predictions reveal differences in the functional architecture of sustained attention and working memory in youth and adults" for consideration as a Research Article at PLOS Biology. Your revised study has now been evaluated by the PLOS Biology editors and the Academic Editor. As the Academic Editor felt comfortable evaluating the revisions, the revised version was not sent back out to the original reviewers. 

I am happy to let you know that we are interested in proceeding further with this work at PLOS Biology, with our Academic Editor feeling that the revisions have nicely addressed the prior reviewer concerns and their own concerns. Our Academic Editor does, however, feel that one additional analysis is needed based on some of the new data that has been added. 

Specifically, in the Supplementary Results section (less strict head-motion exclusion criteria in the ABCD) you mention multiple times that comparing the results in the supplementary section with Tables 1-3 in the main text shows no significant changes in the beta coefficients. However, no statistical values (either frequentist or Bayesian) are presented to support this statement, and we feel that such stats are needed. Our Academic Editor suggests that this null hypothesis would be better supported with Bayesian statistics, and encourages you to perform such analyses to make for a more powerful argument. However, if expertise in Bayesian statistics is not something your lab has access to, we ask that you minimally provide further details based on frequentist statistics.

**IMPORTANT - SUBMITTING YOUR REVISION**

In addition to addressing this statistical issue, when revising the work for resubmission, please address the following points and make sure to carefully address our data and policy issues:

1. Please provide a 'Response to Reviewers' file - this should detail your responses to the additional editorial requests, and indicate the changes made to the manuscript. You should also cite any additional relevant literature that has been published since the original submission and mention any additional citations in your response. 

3. Title change: In order to reach our broad readership, when submitting your revision, we'd ask that you consider a slight title change. We'd suggest:

Differences in the functional brain architecture of sustained attention and working memory in youth and adults

OR (though our editorial view is that the method used isn't necessary in the title):

Differences in the functional brain architecture of sustained attention and working memory in youth and adults revealed by connectome-based comparisons

4. IMPORTANT: Please also make sure to address ALL of the following policies and guidelines while preparing your revision:

*PLOS Data Policy*

Please note that as a condition of publication PLOS' data policy (http://journals.plos.org/plosbiology/s/data-availability) requires that you make open-access available **all** data used to draw the conclusions arrived at in your manuscript. You therefore must include any data used in your manuscript either in appropriate repositories, within the body of the manuscript, or as supporting information (N.B. this includes any numerical values that were used to generate graphs, histograms etc.). For an example see here: http://www.plosbiology.org/article/info%3Adoi%2F10.1371%2Fjournal.pbio.1001908#s5

Note that we **do not** require all raw data. Rather, we ask that all individual quantitative observations that underlie the data summarized in the figures and results of your paper be made available in one of the following forms. The goal is to ensure that our readership has all of the necessary information to recreate the figures in your work. You can provide this information in one of two forms:

2) Deposition in a publicly available repository. Please also provide the accession code or a reviewer link so that we may view your data before publication. Note that we cannot accept sole deposition of data to GitHub or a similar non-static site (e.g. no personal sites and generally no institutional sites; https://journals.plos.org/plosbiology/s/data-availability). Instead, we require deposition to a static site, like Zenodo, FigShare, OSF. GitHub can be used for depositing code however. For your information, note that any data available on a public GitHub site can be directly copied to Zenodo. Once you do this, it will also generate a DOI number that you can provide us with. See the process for doing this here: https://docs.github.com/en/repositories/archiving-a-github-repository/referencing-and-citing-content.

Fig 2,3,4,5,6,7; Fig S1,S3,S4,S5,S6 and final unlabeled graphs in Supp data.

IMPORTANT: Please also ensure that figure legends in your manuscript include information on where the underlying data can be found, and ensure your supplemental data file/s has a legend. (This step is OFTEN forgotten!!)

Please also ensure that your Data Statement in the submission system accurately describes where your data can be found.

*Ethics Statement – Required

While we note that you’ve indicated an ethics statement is required for your work, we did not see this explicitly mentioned in the methods section of your study. It is entirely possible that we missed this, but please do check and confirm that this statement is, indeed, included.

*Blurb – Required

Please provide a blurb which, if the paper is accepted, will be included in our weekly and monthly Electronic Table of Contents (eTOCs), sent out to readers of PLOS Biology. This blurb may also be used to promote your article on social media. The blurb should be about 30-40 words long and is subject to editorial changes. It should, without exaggeration, entice people to read your manuscript, should not be redundant with the title and should not contain acronyms or abbreviations. For examples, view our author guidelines: https://journals.plos.org/plosbiology/s/revising-your-manuscript#loc-blurb

*Published Peer Review*

**Resubmission Checklist**

We expect to receive your revised manuscript within 1 month. Please email us (plosbiology@plos.org) if you have any questions or concerns, or would like to request an extension. At this stage, your manuscript remains formally under active consideration at our journal; please notify us by email if you do not intend to submit a revision so that we withdraw the manuscript.

Sincerely,

Kris

Kris Dickson, Ph.D., (she/her)

Neurosciences Senior Editor/Section Manager

PLOS Biology

kdickson@plos.org

---

## [Editor Report · Decision Letter 3]

30 Nov 2022

Dear Dr Kardan,

Thank you for the submission of your revised Research Article "Differences in the functional brain architecture of sustained attention and working memory in youth and adults" for publication in PLOS Biology. On behalf of my colleagues and the Academic Editor, Roi Cohen Kadosh, I congratulate you on this nice study and am pleased to say that we can in principle accept your manuscript for publication, provided you address any remaining formatting and reporting issues. These will be detailed in an email you should receive within 2-3 business days from our colleagues in the journal operations team; no action is required from you until then. Please note that we will not be able to formally accept your manuscript and schedule it for publication until you have completed any requested changes.

PRESS

We frequently collaborate with press offices. If your institution or institutions have a press office, please notify them about your upcoming paper at this point, to enable them to help maximize its impact. If the press office is planning to promote your findings, we would be grateful if they could coordinate with biologypress@plos.org. If you have previously opted in to the early version process, we ask that you notify us immediately of any press plans so that we may opt out on your behalf.

Sincerely, 

Kris

Kris Dickson, Ph.D., (she/her)

Neurosciences Senior Editor/Section Manager

PLOS Biology

kdickson@plos.org